# Inflammation and Treatment-Resistant Depression from Clinical to Animal Study: A Possible Link?

Lara F. Almutabagani [1], Raghad A. Almanqour [1], Jawza F. Alsabhan [2], Abdulaziz M. Alhossan [2], Maha A. Alamin [3], Haya M. Alrajeh [3], Asma S. Alonazi [3], Ahmed M. El-Malky [4] and Nouf M. Alrasheed [3,*]

1. PharmD. Program, College of Pharmacy, King Saud University, Riyadh P.O. Box 145111, Saudi Arabia
2. Department of Clinical Pharmacy, College of Pharmacy, King Saud University, Riyadh P.O. Box 145111, Saudi Arabia
3. Department of Pharmacology and Toxicology, College of Pharmacy, King Saud University, Riyadh P.O. Box 145111, Saudi Arabia
4. Public Health and Community Medicine, Morbidity and Mortality Review Unit, King Saud University Medical City, Riyadh P.O. Box 145111, Saudi Arabia
* Correspondence: nrasheed@ksu.edu.sa; Tel.: +96-65-0091-8000; Fax: +96-61-1805-4590

**Abstract:** The aim of this study was to investigate the relationship between treatment-resistant depression (TRD) and inflammation in humans and experimental models. For the human study, a retrospective cohort study was conducted with 206 participants; half were on antidepressants for major depressive disorder. The patients were divided into healthy and depressed groups. Inflammation was assessed based on the values of the main inflammatory biomarkers (CRP, WBC and ESR). For the animal experiments, 35 adult male Wistar rats were assigned to stressed and non-stressed groups. Inflammation and stress were induced using lipopolysaccharide and chronic unpredictable mild stress. A 10 mg/kg intraperitoneal injection of fluoxetine (FLX), a known antidepressant, was simultaneously administered daily for 4 weeks. Behavioral tests were performed. The plasma levels of inflammatory and stress biomarkers were measured and were significantly higher in the stressed and non-responsive groups in both studies. This study provides evidence of the link between inflammation and TRD. We further observed a possible link via the Phosphorylated Janus Kinase 2 and Phosphorylated Signal Transducer and Activator of Transcription 3 (P-JAK2/P-STAT3) signaling pathway and found that chronic stress and high inflammation hinder the antidepressant effects of FLX. Thus, non-response to antidepressants could be mitigated by treating inflammation to improve the antidepressant effect in patients with TRD.

**Keywords:** treatment-resistant depression; inflammation; fluoxetine; lipopolysaccharide; chronic unpredictable mild stress; depression

## 1. Introduction

Major Depression Disorder (MDD) is a common psychiatric disorder, characterized by a significantly decreased mood, loss of interest in usual activities and cognitive impairment [1]. Recently, the World Health Organization ranked depression as the third leading cause of disability worldwide and was estimated to rank first by 2030 [2]. The prevalence of depression in Saudi Arabia is estimated to be 4.5% [3,4]. One hypothesis links inflammation to depression and asserts that elevated levels of circulating pro-inflammatory molecules may induce symptoms of depression [5]. Inflammation triggers the immune system to respond to harmful stimuli by releasing pro-inflammatory factors that can lead to physical changes and psychological stress, which can put patients at risk of depression and other illnesses, such as cardiovascular disease, diabetes and cancer [6,7].

Inflammatory biomarkers are correlated with the development of MDD but are not considered to be the direct cause. However, increased concentrations of inflammatory biomarkers have been found in individuals with depression compared to those without

depression. Elevated concentrations of inflammatory markers may indicate a weak antidepressant response [8,9]. However, while approximately half of depressed patients show a sufficient response to antidepressants, around 15% show only a partial response to therapy. Unfortunately, approximately one-third (20 to 35%) of patients with depression are classified as non-responders [10].

Treatment-Resistant Depression (TRD) is defined as the failure to attain sufficient remission of depressive symptoms with adequate medication, dosing and treatment duration. TRD includes (1) failure to respond to one antidepressant medication for at least four weeks and (2) resistance to two or more different classes of adequate antidepressants [11]. Numerous clinical factors linked TRD to inflammation. For example, obesity due to increased inflammatory cytokine levels of transforming growth factor-alpha (TGF-$\alpha$) and interleukin-6 (IL-6), in addition to stress due to the fight-and-flight response, leads to the activation of cortisol and catecholamines [12]. Medical illnesses, such as hypertension, diabetes, cancer and cardiovascular disease, can also result in inflammation [8,12]. Non-response to antidepressants makes treating TRD difficult. Several biomarkers, such as tumor necrosis factor (TNF)-$\alpha$ R1 and C-Reactive Protein (CRP), are related to resistance. TNF-$\alpha$ R1 levels were significantly higher in patients with TRD than in those without TRD [13]. In addition, recent studies have shown that 45% of patients who are resistant to treatment with conventional antidepressant therapy exhibited CRP levels >3 mg/L [14]. Generally, patients with increased levels of inflammatory markers at baseline have been found to be less responsive to antidepressant medications [7,15]. Although the link between inflammation and depression is important and is considered to play the main role in treatment resistance [8], TRD mechanisms warrant further studies and evidence.

The Janus Kinase 2 and Signal Transducer and Activator of Transcription 3 (JAK2/STAT3) signaling pathway plays a pivotal role in immune and inflammatory responses. Dysregulation of the JAK/STAT pathway has been found to be a key factor in various neurodegenerative diseases, and direct evidence from studies in populations with depressive disorders suggests that this pathway may be involved in the pathophysiology of depression [16,17]. However, to our knowledge, no available data suggests the involvement of the JAK/STAT pathway in inflammation-induced TRD.

In this study, several hypotheses were tested to confirm the association between inflammation and TRD. First, we suspected that the depressed group would have higher levels of inflammatory biomarkers than the healthy control group. Second, we examined the preventive role of FLX in environmental and pharmacologically relevant models of depression. Accordingly, we hypothesized that inflammation contributes to TRD. In addition, we observed a potential involvement of the JAK/STAT signaling pathway in our preclinical study settings.

Clinical studies in humans and experiments on animals were conducted to evaluate the research objectives. In the clinical study, we identified whether a relationship exists between antidepressant resistance and inflammation by measuring inflammatory biomarkers CRP, White Blood Cell (WBC) count and Erythrocyte Sedimentation Rate (ESR) and checking the signs that may indicate treatment non-response, such as failure to respond to one or more antidepressants, remaining depressed while receiving medication, being hospitalized several times and committing suicide in the clinical model. The clinical study was based on the outcomes to demonstrate the relationship between TRD and inflammation in human models of depression by measuring inflammatory biomarkers CRP, Interleukin-6 (IL-6) and TNF-$\alpha$ and stress markers (corticosterone). Finally, the mechanisms underlying inflammation-induced TRD were examined using behavioral, biochemical, immunohistochemical and histopathological studies in an animal model of depression.

## 2. Materials and Methods

### 2.1. Clinical Study

### 2.1.1. Study Design and Setting

This retrospective cohort study was conducted at King Khalid University Hospital (KKUH), a tertiary hospital in Riyadh City, which is part of the Ministry of Health in Saudi Arabia. Patients were enrolled by reviewing their electronic medical records, including their patient charts and all their information that met the study inclusion criteria, between January 2015 and December 2020. All study procedures were approved by the research ethics committee (National Committee of Bioethics, Ethics Service and the Institutional Review Board of King Saud University (KSU-IRB); approval No. E-20-4971 (7/12/2020)). All the data were collected solely for the purpose of this study, and patient information was confidential.

### 2.1.2. Participants and Psychiatric Assessment

The inclusion criteria were patients aged 18 years, male or female, with an Axis-1 clinical diagnosis of MDD, and taking antidepressants. The patients were divided into two groups for analyses, with each analysis further divided into two groups, either (1) healthy volunteers vs. patients with depression, or (2) treatment responders vs. non-responders. The patient exclusion criteria were as follows: (1) patients diagnosed with Axis-1 psychiatric disorders other than depression, such as schizophrenia or bipolar disorder; (2) pregnant or breastfeeding during the study period; (3) taking medication that inhibited the immune system, such as corticosteroids and non-steroidal anti-inflammatory drugs; and (4) patients with a known history of alcoholism or drug abuse. In the sensitivity analysis, we also excluded patients with an acute infection, based on the possibility of extremely skewed CRP and WBC levels at the time of blood collection [15,17].

### 2.1.3. Data Sources

Patient data were obtained retrospectively from the electronic medical records of patients who attended psychiatric clinics in KKUH with MDDs under the ICD-10 code numbers F30-F39 and taking antidepressants. We also investigated current and past psychiatric disorders, hospitalizations and medication use, as well as screened for major chronic illnesses, such as cardiovascular disease, diabetes, insulin use, stroke and cancer. We collected information on inflammatory illnesses (rheumatoid arthritis, autoimmune disorders, inflammatory bowel diseases, systemic lupus erythematosus and multiple sclerosis) and medications that might correlate with our results.

### 2.1.4. Measures of Antidepressant Resistance

At the beginning of the study, all demographic details and complete medical histories of the enrolled patients were collected. Current clinical assessments were included by reviewing the physician's notes to rule out the onset of depression. Following this, the staging process of treatment resistance was completed using the "six techniques" for patients with depression. We analyzed antidepressant resistance in the patients by proxy as follows, depending on data availability: (1) history of depression, (2) change of antidepressant in the current year, (3) number of previous depressive episodes, (4) repeated number of hospitalizations for depression and visits committed by the patient since diagnosis of depression, (5) length of depressive episode, and (6) committing suicide [11].

### 2.1.5. Inflammatory Biomarkers

The final patient recruitment step was to identify inflammatory biomarkers, which are correlated with depression. Inflammation was assessed based on the values of the main inflammatory biomarkers (CRP, WBC and ESR) obtained from patient electronic records at KKUH. We analyzed the levels of inflammatory biomarkers as a continuous measure. The values of the biomarkers recorded in the patient files were measured using different methods. CRP levels were measured using an analyzer and WBC count and ESR

were measured using full blood counts. We expected inflammation to be the likely cause of depression [18].

### 2.1.6. Sample Size Calculation

The prevalence of depression in Saudi Arabia is 4.5%, with a population of 1,339,976 [4]. From the total number of patients with depression in Saudi Arabia, we calculated the study sample size with a 99% confidence interval (CI), as we already know the prevalence and population. The total sample size of the screened participants was 650 patients, while those included in both analyses was primarily 206 [19].

### 2.1.7. Statistical Analyses

We conducted two subsequent analyses: (A) inflammation as a continuous variable in the comparison of patients with depression and healthy participants without depression and (B) inflammation as a continuous variable among those who were considered to have potential resistance to antidepressants compared with potential responders to antidepressants.

All data analyses were performed using IBM SPSS Statistics (version 26.0; IBM Corp., Armonk, NY, USA). Descriptive statistics for the study sample are presented as frequencies and relative frequencies (percent) for categorical variables. The mean, standard deviation, median, skewness of the data and interquartile range were used to represent numerical variables. The responder and resistant depression groups were compared using the chi-square test or Fisher's exact test for categorical variables and the independent sample *t* test or Mann–Whitney test for numerical variables. Categorical variables were tested for normal distribution using the Kolmogorov–Smirnov test. Bivariate correlation analysis (Pearson test and Spearman–Rho test) between age, CRP level, WBC count and ESR was performed for the patient groups. The two-tailed *t* test was used to assess the differences between the two groups in variables with normal distributions. The Kruskal-Wallis test for analysis of variance was used for data with abnormal distributions. The significance of the results was set at $p < 0.05$.

We normalized the distribution of the inflammatory markers as needed by natural logarithmic transformation and used binary logistic regression to estimate the odds ratio (OR) and 95% CIs of the inflammatory markers for depressive symptoms. We performed linear regression analysis of depressive symptom scores with adjustment for covariates. To further examine the association between inflammatory proteins and depressive symptoms, multinomial binary logistic analysis was performed using either depressive symptoms or a diagnosis of depression at baseline and inflammatory marker values.

All analyses were controlled for age and gender as continuous variables. We added potential confounders to the basic model to analyze the effects of confounding factors. If this changed the effect estimate significantly, the contribution of each variable was determined individually. In subgroup analyses, we assessed the age- and gender-adjusted associations after the exclusion of participants with acute inflammation or those with and without chronic illnesses, Body Mass Index (BMI) data, current smoking, chronic illnesses and use of any antidepressants, the type of antidepressant e.g., selective serotonin reuptake inhibitor (SSRI), serotonin norepinephrine reuptake inhibitor (SNRI) or other, and a marker of socioeconomic factors, if available, such as the highest education level achieved.

## 2.2. Experimental Study

### 2.2.1. Animals

The experiments were performed using 5–7-week-old adult male Wistar rats with an initial weight of 150–200 g. The rats were obtained from the Animal Care Center at the College of Pharmacy, King Saud University, Riyadh, Saudi Arabia. On arrival, they were housed individually in polypropylene cages with standard laboratory conditions and controlled humidity, a housing temperature of 25 °C ± 1 °C and a 12-h light/dark cycle, with free access to food and water ad libitum for the entire duration of the experiment. The rats were kept under observation for one week to acclimatize them to the laboratory

conditions prior to the experiment. Appropriate measures were taken to minimize pain and discomfort in experimental animals. All experiments were conducted in accordance with the Experimental Animals Ethics Committee Act of King Saud University, Institutional Research Ethics Committee (Ethics Reference No. KSU-SE-20-23 (08/05/2020)).

### 2.2.2. Drugs and Chemicals

The SSRI, FLX and Lipopolysaccharide (LPS) (obtained from *E. coli*, which carries serotype 055: B5) were obtained from Sigma-Aldrich (St. Louis, MO, USA). Enzyme-linked immunosorbent assay (ELISA) kits for corticosterone, IL-6, CRP and TNF-$\alpha$, were purchased from Abcam Biotechnology, Inc. (Cambridge, UK). All the drugs were freshly prepared and injected intraperitoneally (i. p.) at a final injection volume of 5 mL/kg. The injections were administered between 8:00 AM and 10:00 AM, irrespective of the stress schedule. Rabbit polyclonal anti-phospho-JAK2 and goat anti-phospho-STAT3 (pSTAT3) were supplied by Santa Cruz Biotechnology, Inc. (Dallas, TX, USA). All the other chemicals were of analytical grade and were obtained from Sigma-Aldrich or Cell Signaling Technology (Beverly, MA, USA).

### 2.2.3. Experimental Design

The entire experiment was conducted over the course of 4 weeks. Thirty-five rats were randomly assigned to seven different groups, each containing five rats.

(Group 1): Unstressed rats treated with normal saline (0.9% NaCl).

(Group 2): Stressed rats with chronic unpredictable mild stress (CUMS) treated with normal saline [20].

(Group 3): Non-stressed rats treated with FLX, 10 mg/kg/day, i.p. [18].

(Group 4): Stressed rats treated with FLX.

(Group 5): Non-stressed rats treated with 100 µg/kg LPS i.p. [21].

(Group 6): Stressed rats treated with LPS before CUMS exposure.

(Group 7): Stressed rats treated with LPS prior to CUMS exposure and FLX treatment.

### 2.2.4. Experimental Procedures

Based on the experimental design, inflammation and depression were induced in rats according to their grouping to determine whether the final goal of treatment resistance was achieved. At the end of the 4-week stress period, behavioral tests were performed and treatment resistance in depressive rats was determined and measured. Figure 1 demonstrates the experimental design.

#### Induction of Inflammation by LPS

In a rat model, inflammation was induced by administering LPS for 4 weeks to mimic inflammatory activity. An LPS dose of 100 µg/kg, i.p. was administered once daily to rats prior to the introduction of stressors [21].

#### Induction of Depression by CUMS

The CUMS protocol is an established translationally relevant model for inducing behavioral symptoms commonly associated with clinical depression in rodents, such as anhedonia, altered grooming behavior and learned helplessness [22]. We created a chronic unpredictable mild stressful environment using stress and diet to induce antidepressant resistance. The rats were subjected to various CUMS treatments for 4 weeks. Alterations in bedding (repeated changing position and removal of sawdust and damp sawdust), cage tilting (at an angle of 45°), intermittent noise, and water deprivation, followed by exposure to an empty water bottle and alteration in the light/dark cycle, were applied as stressors. This series of stressors was applied for one week and repeated over a course of four weeks [20,21]. A successful sucrose preference test (SPT) confirmed the development of depression in rats [23]. After the 4-week stress exposure, sucrose preference was signifi-

cantly reduced in the CUMS group, compared to that in the control group [24]. Behavioral tests were performed on the day after the last stressor.

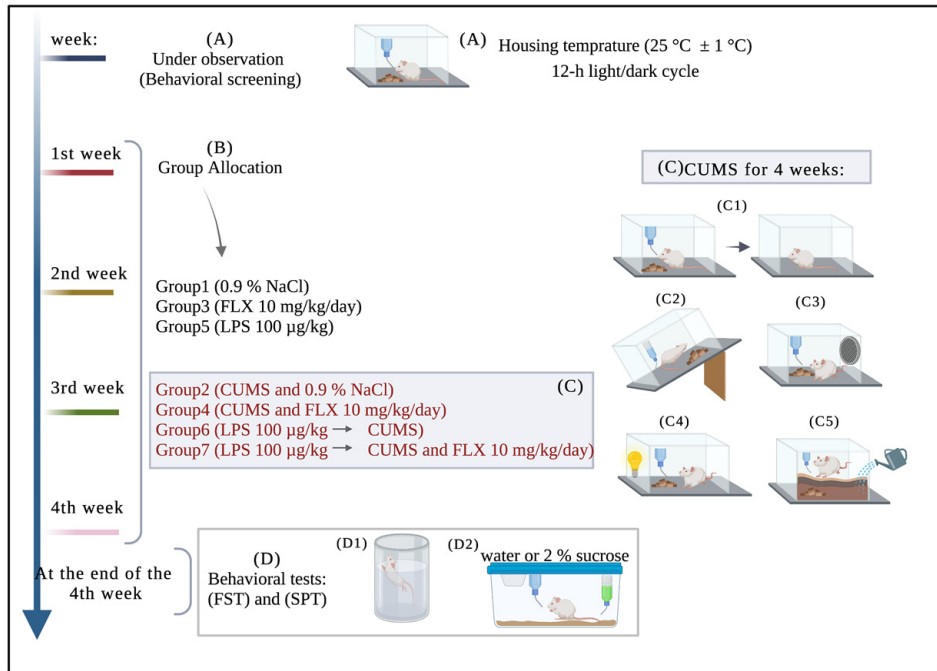

**Figure 1.** Experimental Protocol: (Week 0) the rats were under observation. Thereafter, during the four weeks (Week 1 to Week 4) rats were divided into stressed and non-stressed groups and chronic unpredictable mild stress protocol was applied: (C1) food and water deprivation, (C2) cage tilting, (C3) intermittent noise, (C4) alteration of light/dark cycle, (C5) bedding alteration and wetting. At the end of the 4th week behavioral tests were applied: Forced Swim Test and Sucrose Preference Test. NaCl: sodium chloride (normal saline); CUMS: chronic unpredictable mild stress; FLX: fluoxetine; LPS: lipopolysaccharide; FST: forced swim test; SPT: sucrose preference test.

Behavioral Tests

Unstressed rats were isolated 3 h before behavioral testing. The rats were habituated to the testing room for 30 min before the behavioral analysis test was started. The forced swimming test (FST) and SPT were conducted. All the tests were conducted between 10:00 AM and 4:00 PM.

Forced Swimming Test

The FST, which is one of the most commonly used experiments to study depression-like behaviors in rodents, was performed as described by Porsolt et al., with minor modifications [25,26]. The rats were individually placed in a plastic cylinder (height: 45 cm; diameter: 20 cm) filled to a 30-cm depth and maintained at $25 \pm 1\ ^\circ C$. The rats were forced to swim for 15 min. They were then dried and returned to their cages. After 24 h, all the rats were exposed to the FST for 5 min. In the second session, treatments in the various groups were administered 1 h prior to the FST. The cylinder was freshly cleaned and disinfected prior to the FST. Clean water was used for each behavioral trial.

Sucrose Preference Test

The SPT was performed to assess anhedonia (decreased ability to feel pleasure), which is one of the most common symptoms of major depression. First, the rats were allowed to adapt to the sucrose solution (2% $w/v$) by placing two bottles of sucrose solution in each cage for 24 h. Second, the rats were housed individually and one bottle of sucrose solution, each containing a specific volume (200 mL), was replaced with water. Sixty minutes later,

sucrose consumption was measured as follows (sucrose preference = V (sucrose solution) / [V (sucrose solution) + V (water]) × 100%) [27].

### 2.2.5. Preparation of the Blood and Brain Samples

At the end of the experiment, all the rats were weighed, anesthetized using a gradually increasing concentration of carbon dioxide ($CO_2$) and then sacrificed by decapitation. Blood samples were collected and serum was separated from aliquots of the blood samples to determine the levels of inflammatory cytokines and perform other biochemical analyses. The entire brain was quickly removed, rinsed with ice-cold phosphate-buffered saline (PBS) and weighed. The hippocampi were then isolated, frozen in liquid nitrogen and stored at −80°C for further biochemical analysis. Hippocampal samples were homogenized in cold PBS (10% $w/v$) and a clear homogenate was collected to assay the levels of cytokines (IL-6, TNF-$\alpha$ and CRP) using ELISA kits according to the manufacturer's instructions. Brain tissues were dissected, collected and fixed in neutral buffered formalin (4%) for histological and immunohistochemical analyses [28].

### 2.2.6. Assessment of Inflammatory Biomarkers

Cytokines regulate immune responses and inflammatory processes. TNF, interferons, interleukins and colony stimulatory factors belong to a class of cytokines, and their activation leads to the activation of CRP. All of these cytokines indicate the presence of inflammation. To determine serum levels of IL-6, TNF-$\alpha$ and CRP, a specific rat ELISA kit was used. Postmortem blood samples were collected via cardiac puncture and centrifuged at 3300 rpm for 10 min at room temperature. Plasma aliquots were stored at −80 °C until ELISA analysis. Commercially available sandwich ELISA kits were used to analyze CRP, IL-6 and TNF$\alpha$ levels in accordance with the manufacturer's instructions and the samples were analyzed in duplicate. Owing to assay size limitations, the samples were spiked with IL-6 (37.5 μL at 1500 pg. μg/mL standard) and TNF-$\alpha$ (25 μL of 2000 pg. μg/mL standard) to improve the assay sensitivity. Protein absorbance was measured using a VersaMax tunable microplate reader (SoftMax Pro 5.4.1 software (Molecular Devices, Sunnyvale, CA, USA) set at a wavelength of 470 nm with 570 nm correction, in accordance with the manufacturer's instructions [29].

### 2.2.7. Measurement of Plasma Corticosterone

Accurate measurement of plasma corticosterone concentration, the primary stress hormone in rodents, is an important step in detecting the stress response in experimental animals. The plasma serum levels of corticosterone were assayed using ELISA kits (Abcam), which quantified the corticosterone concentration in a sample based on competing interactions of either endogenous or enzyme-linked antigens with limited amounts of antibody [30]. The assay sample and buffer were incubated with a corticosterone-horseradish peroxidase (HRP) conjugate in a precoated plate for 1 h. The wells were decanted, washed five times and incubated with the HRP substrate. A stop solution was added and the intensity of the yellow color formed was measured spectrophotometrically at 450 nm using a microplate reader. A standard curve was plotted and the corticosterone concentration in each sample was interpolated from this standard curve according to the manufacturer's instructions [16].

### 2.2.8. Histological Examination

Histopathological assessments were performed on the brains of randomly selected rats from each group. After induction of animal anesthesia with $CO_2$, the brain was excised, rinsed with PBS and immediately placed in 4% paraformaldehyde solution for 24 h. Paraffin-embedded brain specimen sections (from the hippocampus) were stained with hematoxylin and eosin for morphological examination. Paraffin-embedded sections (2-μm slices) were incubated overnight with rabbit anti-rat antibody (1 × 400) and exposed

to the secondary antibody. The images were prepared by investigators specializing in this field [31].

### 2.2.9. Immunohistochemistry for Assessment of Expression of JAK2/STAT3

Immunohistochemistry was used to detect the expression of JAK2 and STAT3 proteins in brain tissue. After rat brains were removed, serial coronal sections of the entire hippocampus were cut using a sliding microtome. The sections from each brain were embedded in paraffin sections, de-waxed, hydrated and placed in an EDTA-containing antigen repair buffer (pH 9.0) to detect the immunostaining of the brain sections. Antigen retrieval was performed in a microwave oven. The slices were placed in a 3% hydrogen peroxide solution to block endogenous peroxidase and then blocked with a 5% bovine serum albumin solution at room temperature for 30 min. The sections were then incubated with goat polyclonal anti-phospho-JAK2 (Tyr 1007/1008) (SC-21870 from Santa Cruz Biotechnology, Inc.) and mouse monoclonal anti-phospho-STAT3 (B-7; Tyr 705) (SC-8059 from Santa Cruz Biotechnology, Inc.) at 4 °C overnight, followed by incubation with secondary antibodies for 50 min at room temperature. The slices were washed for 15 min and DAB color-developing solutions were used. The slides were observed under a microscope and the time was recorded. Finally, positive staining was indicated by the spots. The cells were then washed with tap water. Each slice was dehydrated and sealed after the nuclei were stained with hematoxylin.

### 2.2.10. Statistical Analysis

The mean and standard error of the mean (SEM) were used to express the data. Statistical analyses were performed using GraphPad Prism 6 (GraphPad Software, San Diego, CA, USA). Intergroup significance was evaluated using one-way analysis of variance (ANOVA) followed by a Tukey–Kramer comparison post hoc analysis. $p$ values $< 0.05$ were considered significant.

## 3. Results

### 3.1. Clinical Results

#### 3.1.1. Study Population, Demographic Characteristics and Clinical Data

The extracted data were obtained from January 2015 to December 2020. Of the 650 participants, 206 met the study criteria and were enrolled in the assessment. A total of 103 patients were healthy controls and 103 were diagnosed with MDD. Thirty-nine patients from the depressed group were considered treatment responders and 40 were treatment-resistant. In total, 444 participants were excluded from the initial cohort because they met the exclusion criteria (Figure 2).

The study population size of each group and their demographic and clinical characteristics are summarized in Supplementary Tables S1 and S2, respectively. Age, BMI and laboratory results of the patients are presented as a mean, standard deviation, median and interquartile range (Q1, Q3). Comparisons of the groups in each analysis were performed using the chi-square test or Fisher's exact test for categorical variables and the independent sample $t$ test for age and BMI.

In the first analysis (Table 1: patients with depression vs. healthy controls), demographic characteristics did not differ significantly between the groups, as shown in Table S1. The only variable that showed a statistically significant difference between the two groups was education, as 75.9% ($n = 22$) of the healthy group was educated (had advanced knowledge and skills by having their high school or bachelor degree), compared to 23.1% ($n = 21$) only in the depressed group. A comparison of the laboratory test results between the two groups was performed using the independent sample $t$ test.

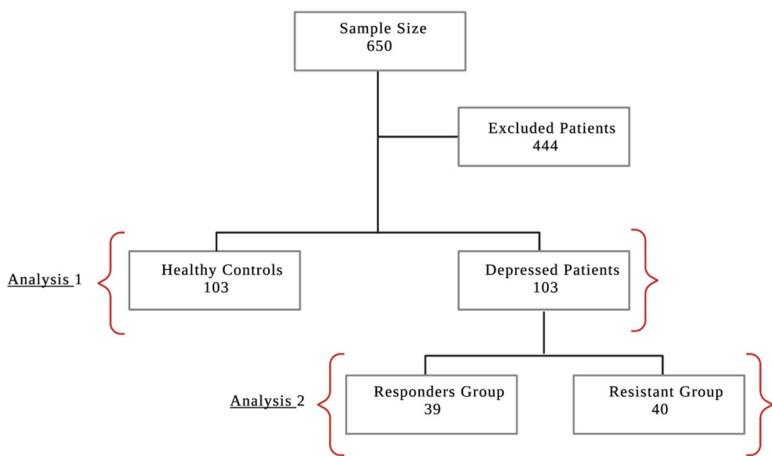

**Figure 2.** Study Sample Size and Randomization of Patients Flow: The total calculated sample size (*n* = 650). (*n* = 444) patients were excluded. The rest of the patients (*n* = 206) that met our inclusion criteria were divided into groups (Healthy controls vs. Depressed patients) which represents Analysis 1. The depressed group were divided into subgroups which represents Analysis 2 (Responder patients vs. Resistant patients).

**Table 1.** Comparison of the Depressed Group vs. Healthy Group in the Demographic and Clinical Data.

| Associated Factors | | Group | | *p*-Value |
|---|---|---|---|---|
| | | Healthy | Depression | |
| Gender | Female | 62 60.2% | 71 68.9% | 0.190 |
| | Male | 41 39.8% | 32 31.1% | |
| Education | No | 7 24.1% | 70 76.9% | <0.001 |
| | Yes | 22 75.9% | 21 23.1% | |
| Smoking | No | 95 96.9% | 95 94.1% | 0.498 |
| | Yes | 3 3.1% | 6 5.9% | |
| Chronic Disease | No | 42 40.8% | 33 32.0% | 0.193 |
| | Yes | 61 59.2% | 70 68.0% | |
| Age Mean (SD) | | 51.9(18.7) | 52.6(16.6) | 0.777 |
| BMI Mean (SD) | | 29.9(7.2) | 30.8(7.8) | 0.378 |

BMI: body mass index; SD: standard deviation. All variables showed non-significant differences except for education, which differed significantly between the healthy volunteers and the depressed patients with a *p* value < 0.001.

In the second analysis (Table S2: treatment responders vs. treatment-resistant), the variables used to measure and differentiate treatment-resistant patients from treatment-responsive patients showed a statistically significant difference. These variables were used to rule out patients who had treatment-resistant depression, including taking antidepressants, improvement with medications, suicide attempts, repeated hospital visits, changes in antidepressants in the current year and adherence. A total of 81.8% (*n* = 33) of the respondents were taking antidepressant drugs, compared to 97.5% (*n* = 39) in the resistant group. All patients in the respondent group (100%, *n* = 39) showed improvement with medication, compared to 12.5% (*n* = 5) of the resistant group. Twenty percent (*n* = 8) of the resistant group reported suicide attempts, compared to 0% (*n* = 0) of the respondent group. Repeated visits were observed in 77.5% (*n*= 31) of the resistant group and only 27.3% (*n* = 6)

of the respondent group. A change in AD treatment in the current year was observed in 55% (*n* = 22) of the resistant group and 4.5% (*n* = 1) of the respondent group. A total of 81.8% (*n* = 4) of the responders adhered to the antidepressant medications, whereas 92.5% (*n* = 37) of the resistant group did not adhere to the therapeutic protocol.

The rest of the variables did not show statistically significant differences between both groups.

The other variables, education, gender, smoking, inflammatory disease, chronic disease, hospitalization, family history, age and BMI, did not show any significant differences between the two groups. A comparison of the laboratory test results in the second analysis was performed using the Mann–Whitney test for all variables except Low Density Lipoprotein (LDL) and cholesterol, where the independent *t* test was used.

### 3.1.2. Evaluation of Inflammatory Biomarkers (CRP, ESR, WBC)

Analysis 1: Healthy Group vs. Depressed Group

As shown in Figure 3, the mean high-sensitivity CRP concentrations were significantly higher in all patients with MDD, compared with healthy controls ($57.1 \pm 82.7$ mg/L vs. $6.5 \pm 11.5$ mg/L, $p < 0.001$). Moreover, the results showed a statistically significant difference in the ESR and WBC count, which were higher in the depressed group than in the healthy controls. The ESR level in the depressed group vs. the healthy controls was $44.7 \pm 31.3$ mg/L vs. $30.6 \pm 22.7$ mg/L; $p = 0.004$. The level of WBC in the depressed group vs. the healthy controls was ($8.9 \pm 4.8$ mg/L vs. $7.5 \pm 2.4$ mg/L; $p = 0.007$). This suggests that patients with depression have higher levels of inflammatory biomarkers than healthy participants.

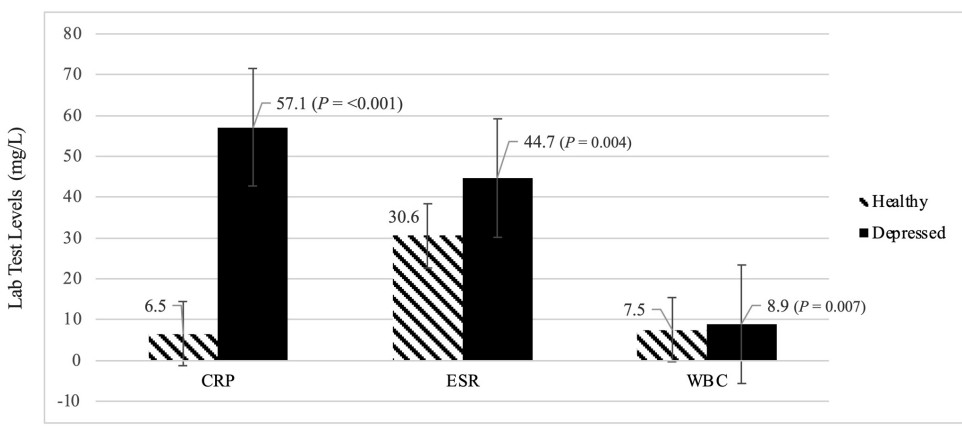

**Figure 3.** Comparison of the Mean Lab Test Results between the Healthy Group and the Depressed Group. CRP: C-reactive protein; ESR: erythrocyte sedimentation rate; WBC: white blood cells.

Analysis 2: Treatment Responders Group vs. Treatment Resistant Group

As shown in Figure 4, the mean value of high-sensitivity CRP concentration was significantly higher in patients who had treatment-resistant depression, compared to the treatment-responsive patients ($61.0 \pm 97.0$ mg/L vs. $11.08 \pm 7.7$ mg/L; $p = 0.031$). The ESR showed statistically significant differences between the treatment-resistant group and the treatment-responsive group. The ESR level was significantly higher in the treatment-resistant patients, compared to the treatment-responders ($50.7 \pm 31.4$ mg/L vs. $27.2 \pm 12.4$ mg/L; $p = 0.048$). Although the WBC levels did not differ significantly between the groups ($p = 0.174$), the significant difference in the CRP and ESR biomarker levels between the resistant and responder groups provides strong evidence of a relationship between inflammation and treatment-resistant depression.

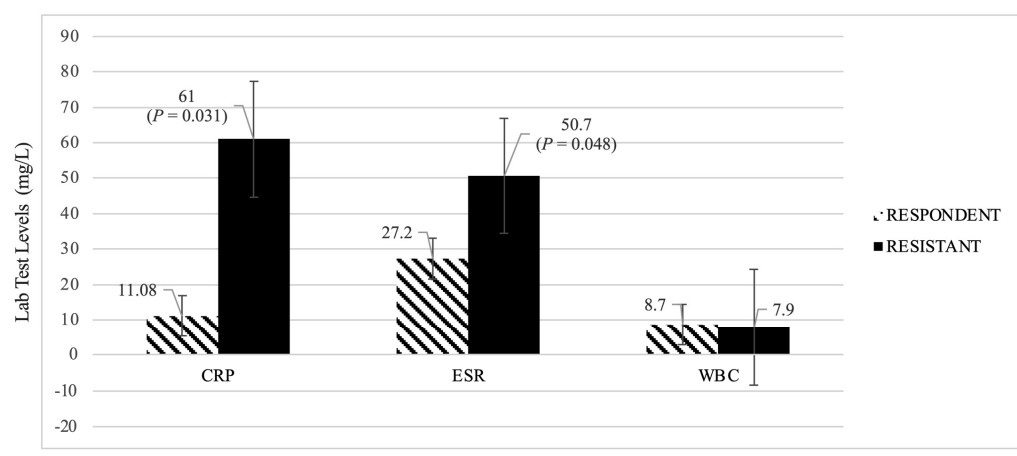

**Figure 4.** Comparison of the Mean Lab Test Results between the Respondent Group and the Resistant Group. CRP: C-reactive protein; ESR: erythrocyte sedimentation rate; WBC: white blood cells.

*3.2. Experimental Results*

3.2.1. Fluoxetine-resistant Depression Model and Behavior Evaluation

The Sucrose Preference Test (SPT)

Figure 5 shows the mean of the SPT results of the rats at the end of the stress phase of the experiment (week 4). After four weeks of stress exposure, sucrose preference was non-significantly reduced in rats treated with LPS and exposed to CUMS, compared to CUMS (Group 2). At week 4, stress exposure (CUMS) induced a significant decrease in sucrose preference in Group 2 compared to that in the control Group 1 (69.6 ± 10.10% vs. 96 ± 1.53%; $p < 0.05$). In FLX treatment, there was a non-significantly increased sucrose preference in rats exposed to CUMS (Group 4), compared to rats exposed to CUMS only without treatment (Group 2: 84.716 ± 8.29% vs. 69.6 ± 10.10%). The sucrose preference percentage was lower in the FLX-treated Group 7 than in FLX-treated Group 4 (48.483 ± 15.52% vs. 84.716 ± 8.29%; $p < 0.01$). Sucrose preference was lowest in the group exposed to CUMS + LPS (Group 7), because LPS attenuated the effect of FLX. In the control group, FLX did not affect sucrose preference.

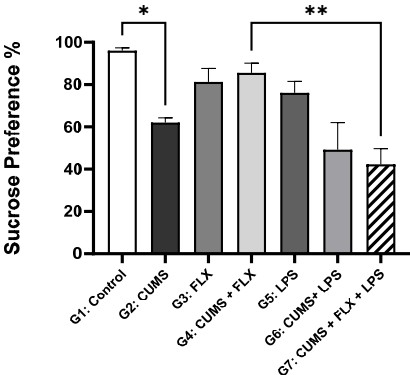

**Figure 5.** Effects of Lipopolysaccharide on Sucrose Consumption in Fluoxetine Treatment Resistant Depressive Rats: Data are expressed as the means ± SEM, *n* = 5. Group comparisons were performed using one-way ANOVA, followed by a Turkey–Kramer post hoc test; * $p < 0.05$ compared to control non-stressful group (group 1); ** $p < 0.01$ compared to FLX-treated stressful group (Group 4). CUMS: chronic unpredictable mild stress; FLX: fluoxetine; LPS: lipopolysaccharide.

The Forced Swimming Test

Figure 6 shows the mean immobility time displayed by the rats in the FST, in which they developed an immobile posture when placed in an inescapable cylinder filled with

water. In the FST, the depressed rats that were exposed to CUMS (Group 2) spent a longer time immobile than the control group (Group 1; 64.3 ± 12.86 s vs. 41 ± 4.36 s, $p < 0.01$). The immobility time was non-significantly decreased by FLX in the CUMS group (Group 4), compared with the control (Group 1) (9.6 ± 4.84 s vs. 41 ± 4.36 s). Moreover, the CUMS-untreated rats (Group 2) showed a significantly higher immobility time than the FLX-treated CUMS group (Group 4; 64.3 ± 12.86 s vs. 9.6 ± 4.84 s; $p < 0.01$). In the FLX-treated (CUMS + LPS) Group 7, the fluoxetine effect was attenuated, since the rats had a high immobility time, compared with the FLX-treated CUMS Group 4 (50.6 ± 15.5 s vs. 9.6 ± 4.84 s; $p < 0.001$). The FLX-treated (CUMS + LPS) rats in Group 7 showed no significant difference in the duration of immobility time, compared with the CUMS-untreated Group 2 (50.6 ± 15.5 s vs. 64.3 ± 12.86 s), since LPS masked the effect of FLX in the resistance group (Group 7).

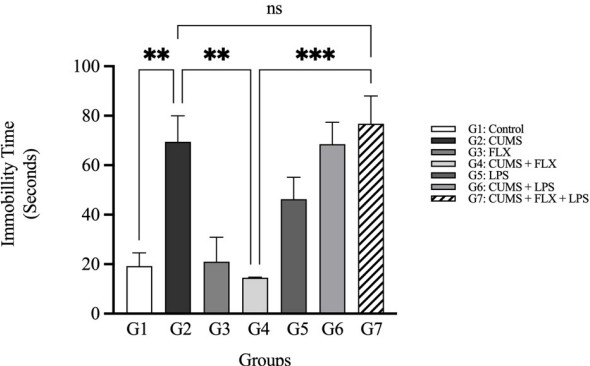

**Figure 6.** Effect of Lipopolysaccharide on Latency and Duration of Immobility in Fluoxetine Treatment Resistant Depressive Rats Exposed to Forced Swimming Test (Seconds): Data are expressed as the means ± SEM; *n* = 5. Group comparisons were performed using one-way ANOVA followed by a Turkey–Kramer post hoc test; ** $p < 0.01$; *** $p < 0.001$. CUMS: chronic unpredictable mild stress; FLX: fluoxetine; LPS: lipopolysaccharide; ns: non-significant.

### 3.2.2. Fluoxetine-resistant Depression Model and Evaluation of Stress Markers
Corticosterone

Figure 7 shows the serum corticosterone levels of the various treatment groups. The non-stressed + LPS Group 5 had significantly higher corticosterone levels than the control Group 1 (51.77 ± 3.09 vs. 44.29 ± 2.60 ng/mL; $p < 0.01$). The corticosterone level increased in untreated (CUMS + LPS) rats (Group 6) when compared with non-stressed + LPS rats in Group 5 (52.09 ± 0.85 vs. 51.77 ± 3.09 ng/mL). Inversely, the effect of FLX on serum corticosterone levels was attenuated in the FLX-treated CUMS + LPS rats in Group 7, where the corticosterone serum level was higher than that of stressed (CUMS) rats in Group 2 (54.40 ± 1.02 vs. 49.09 ± 1.22 ng/mL) and FLX-treated CUMS rats in Group 4 (54.40 ± 1.02 vs. 45.84 ± 2.10 ng/mL).

### 3.2.3. Fluoxetine-resistant Depression Model and Evaluation of Inflammatory Markers
C-Reactive Protein Plasma Levels

As shown in Figure 8 (panel A), a statistical analysis revealed a significant difference in serum CRP levels between the different groups. The serum CRP level was elevated in stressed (CUMS) rats in Group 2, compared to the control Group 1 (0.226 ± 0.08 vs. 0.126 ± 0.05 pg/mL). The CRP level was higher in the FLX-treated (CUMS +LPS) rats (Group 7) who were exposed to inflammation by LPS in comparison to the FLX-untreated stressed (CUMS) rats (Group 2) who were not exposed to inflammation (0.46 ± 0.03 vs. 0.226 ± 0.07 pg/mL). Similarly, the treatment-resistant (FLX + CUMS + LPS) rats in Group 7 had significantly higher serum CRP levels than the fluoxetine treated (CUMS) rats (Group 4; 0.46 ± 0.03 vs. 0.049 ± 0.03 pg/mL; $p < 0.01$).

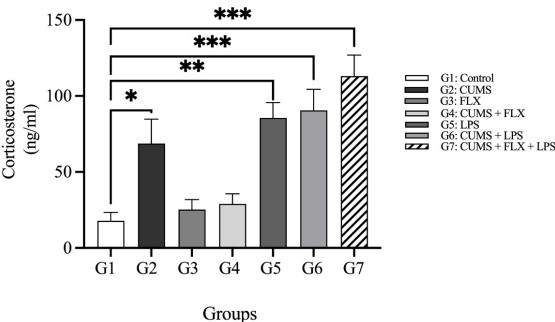

**Figure 7.** Effect of Lipopolysaccharide on Serum Corticosterone Levels in Fluoxetine Treatment Resistant Depressive Rats: Schematic represents corticosterone level (ng/mL) in animal groups exposed to various treatment. Data are expressed as the means ± SEM; *n* = 5. Group comparisons were performed using one-way ANOVA followed by a Turkey–Kramer post hoc test; * *p* < 0.05; ** *p* < 0.01; *** *p* < 0.001. CUMS: chronic unpredictable mild stress; FLX: fluoxetine; LPS: lipopolysaccharide.

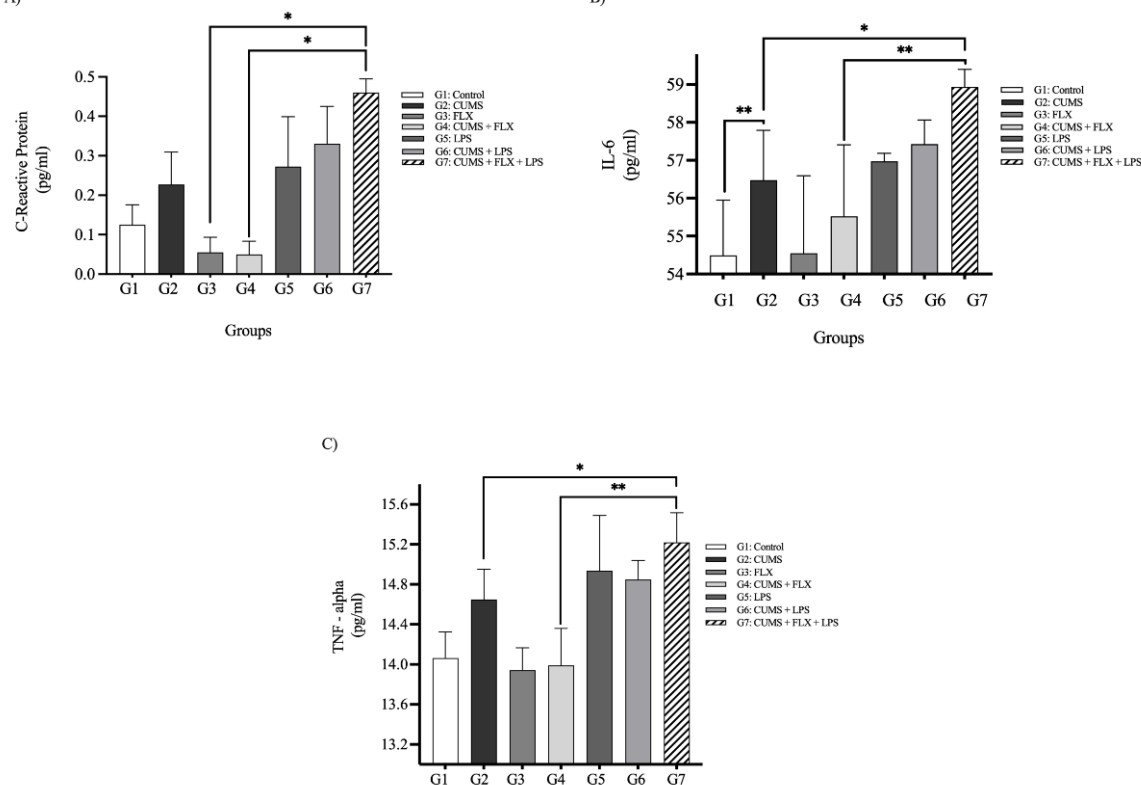

**Figure 8.** (**A–C**) Effects of Lipopolysaccharide on Cytokines Markers (IL-6, TNF-α, and C-Reactive Protein (CPR) Serum Levels in Fluoxetine-Resistant Depression Rats Exposed to CUMS model: Data are expressed as the means ± SEM; *n* = 5. Group comparisons were performed using one-way ANOVA followed by a Turkey-Kramer post hoc test; * *p* < 0.05; ** *p* < 0.01;. CUMS: chronic unpredictable mild stress: FLX: fluoxetine; LPS: lipopolysaccharide; IL-6: interlukien-6; TNF-alpha: tumor necrosis factor -alpha.

Inflammatory Cytokines (IL-6, TNF-α)

As shown in Figure 8 (panel B and panel C), the serum levels of IL-6 and TNF-α were higher in the stressed (CUMS) rats (Group 2), when compared to the control Group 1: IL-6: (56.47 ± 1.31 vs. 54.49 ± 1.45 pg/mL; *p* < 0.01); TNF-α: (14.65 ± 0.30 vs. 14.06 ± 0.26 pg/mL; *p* < 0.05). LPS increased the levels of inflammatory markers (IL-6 and TNF-α) in the FLX-untreated (CUMS + LPS) rats (Group 6), compared to

the control rats that were not exposed to the inflammatory agent (LPS; Group 1); IL-6: (57.42 ± 0.64 vs. 54.49 ± 1.45 pg/mL); TNF-$\alpha$ (14.85 ± 0.18 vs. 14.06 ± 0.26 pg/mL). The serum IL-6 and TNF-$\alpha$ levels were significantly higher in the FLX-nontreated stressed (CUMS) rats (Group 2), and in the treated stressed rats who were exposed to inflammation (FLX + CUMS + LPS; Group 7); IL-6 (56.47 ± 1.31 vs. 58.93 ± 0.46 pg/mL; $p < 0.05$); TNF-$\alpha$ (14.65 ± 0.30 vs. 15.22 ± 0.29 pg/mL), respectively.

After four weeks of FLX treatment, the inflammatory cytokines decreased in the FLX-treated (CUMS) rats (Group 4), but not in the FLX-treated (CUMS +LPS) rats (Group 7), where the measured serum levels of the inflammatory biomarkers of both groups were IL-6: (55.52 ± 1.88 vs. 58.93 ± 0.46 pg/mL; $p < 0.01$) and TNF-$\alpha$ (13.99 ± 0.37 vs. 15.22 ± 0.29 pg/mL; $p < 0.001$), respectively. FLX failed to attenuate the serum levels of inflammatory biomarkers, particularly IL-6 and TNF-$\alpha$, in FLX-treated (LPS + CUMS) rats (Group 7), due to exposure to the inflammatory agent (LPS), which causes treatment resistance.

### 3.2.4. Fluoxetine-resistant Depression Model and Histological Examination

Figure 9 shows an examination of the brain sections of the hippocampus in different groups. The control group showed a normal neuronal population in the hippocampus and a normal pyramidal cell distribution without pyknosis in the cerebral cortex (Panel A). The stressed group showed severely injured neuronal tissues in the form of degeneration, neuronal nucleus shrinkage, vacuolization of pyramidal cells in the hippocampus and a severely irregular appearance of Purkinje cells, with pyknotic nuclei distribution in the cerebral cortex (Panel B).

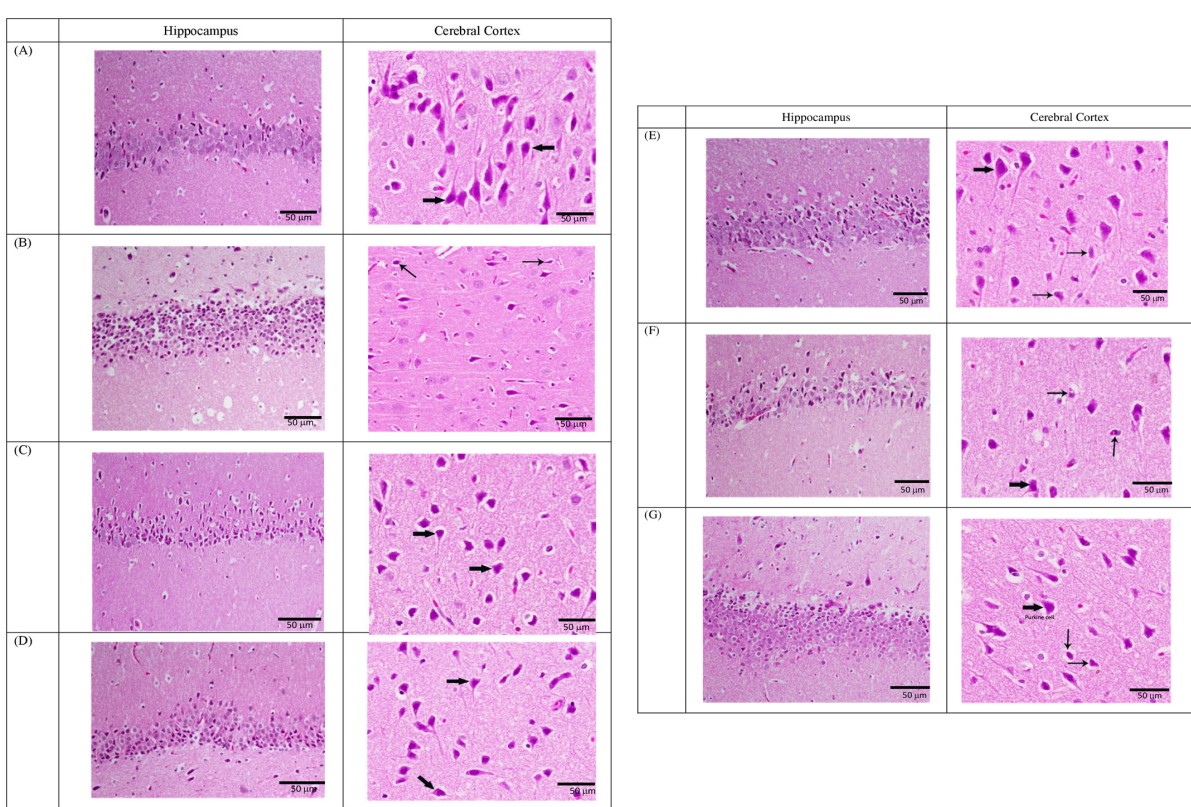

**Figure 9.** Histology Examination of the Brain Sections (Hippocampus and Cerebral Cortex) of the Different Rat Groups: (**A**) control group, (**B**) stressed group, (**C**) fluoxetine-treated non-stressed rats, (**D**) fluoxetine-treated stressed rats, (**E**) pre-injected LPS non-stressed rats, (**F**) pre-injected LPS stressed rats, (**G**) pre-treated fluoxetine and LPS stressed rats (Resistant Group). Each image is a representative image of the brains of five rats per group.

In FLX-treated non-stressed rats, a histological examination of the brain sections did not show any changes in the neuronal tissues and pyramidal cell distribution in the cerebral cortexes or hippocampi (Panel C). The FLX-treated stressed rats showed an improvement in neuronal shape with mild vacuoles or insignificant changes in the hippocampus, and a decreased incidence of neuropathological lesions with an improvement in cerebellar shape in the cerebral cortex (Panel D). In the pre-injected lipopolysaccharide non-stressed group, the brain structure revealed a degeneration of neurons with vacuolization and pyknosis in the hippocampus, and degeneration of neurons with vacuolization organization of the pyramidal cells and cerebellum shape in the cerebral cortex (Panel E). However, pre-injected lipopolysaccharide-stressed rats showed a shrinkage with pyknotic nuclei in some pyramidal cells in the hippocampus. The cerebral cortex showed less degeneration with vacuolization and pyknosis (Panel F). In the resistant group, pretreated FLX-and lipopolysaccharide-stressed rats showed normal neuronal tissues with a few layers of large pyramidal cell regions in the hippocampus, and almost showed an ameliorative effect near a normal pyramidal cell distribution in the cerebral cortex (Panel G).

### 3.2.5. Immunohistochemical Study of the Role of Phosphorylated Janus Kinase 2 (P-JAK2) and Phosphorylated Signal Transducer and Activator of Transcription 3 (P-STAT3) Pathway in FLX-resistant Depression Rats

To further explore whether inflammation contributes to FLX treatment resistance in rats by modulating the JAK2/STAT3 signaling pathway (Figure 10A,B), we detected the expression of JAK2 and STAT3 proteins. Our observation indicated a possible trend the phosphorylation level in the FLX-treated rats who get exposed to (stress and inflammation) together.

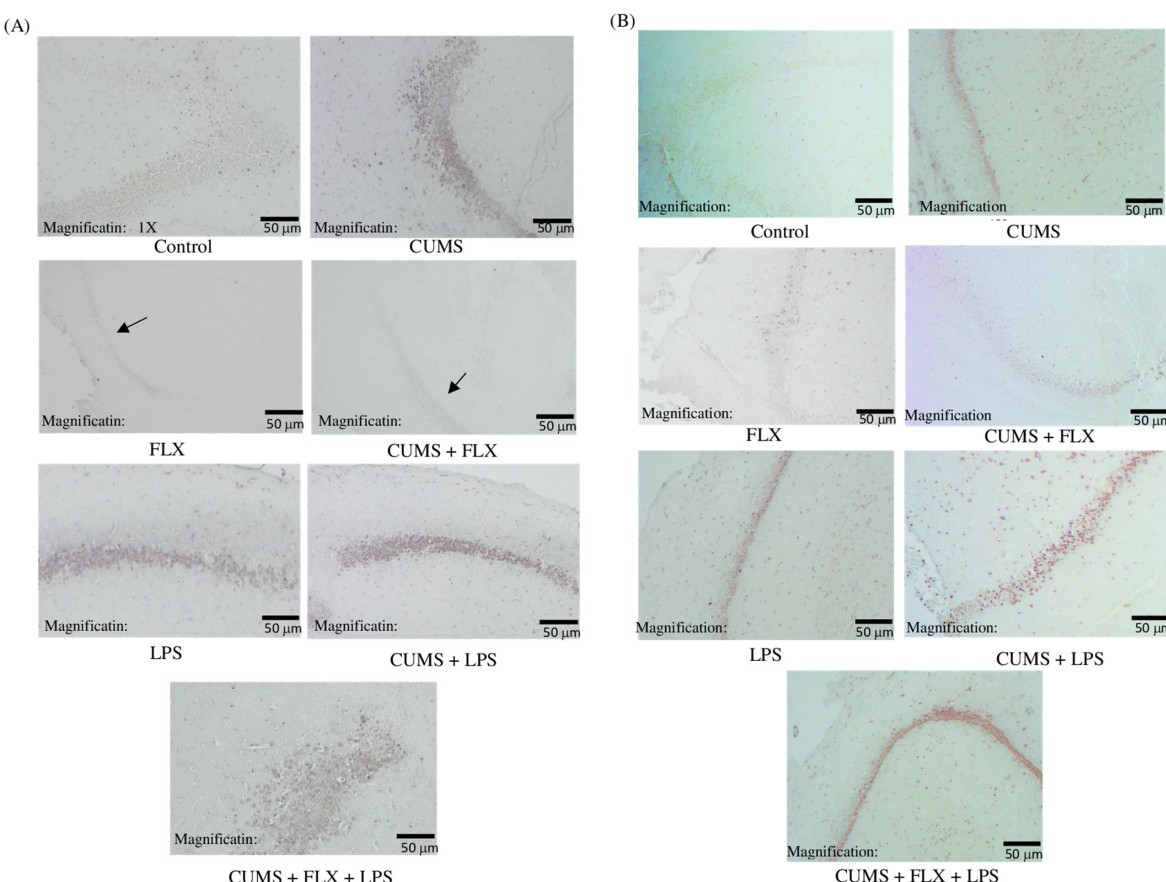

**Figure 10.** (**A**,**B**) Effects of Lipopolysaccharide on P-JAK2 and P-STAT3 Pathway in Fluoxetine-Resistant Depression Rats; CUMS: chronic unpredictable mild stress: FLX, fluoxetine; LPS: lipopolysaccharide. Each image is a representative image of the brains of five rats per group.

## 4. Discussion

TRD remains a common condition, accounting for approximately 30% of all patients with depression [32]. Several pieces of evidence suggest that inflammation plays a major role in the pathogenesis of multiple neurological disorders, including depression [33–35]. Various studies have investigated the relationship between inflammation and the antidepressant non-response, suggesting that inflammatory processes may influence the development of TRD [21]. Moreover, recent studies have indicated that treatment resistance may be associated with increased inflammation. Plasma concentrations of stress and inflammatory biomarkers (cortisol, CRP, IL-6 and TNF-$\alpha$) have been found to be increased in patients with major depressive disorder who have a history of treatment non-response, compared to treatment-responsive patients [14,15,36]. Similarly, animal studies have shown that rats exposed to CUMS and inflammation induced by an inflammatory agent (LPS) had higher levels of brain cytokines (TNF$\alpha$, IL-6, IL-1$\beta$, IFN-alpha and CRP), which affects the antidepressant treatment response more than other groups [14,21]. Thus, we studied this relationship using male rats because the sex-related differences in antidepressant pharmacokinetics and pharmacodynamic suggest that treatment response differ between the sexes. Moreover, the presence of estrogen in females of reproductive age may interfere with the mechanism of action of multiple antidepressant drugs [37,38].

This project investigated the link between inflammation and TRD through clinical human and animal studies and is the first in the field of neuropsychiatric research to investigate the link between inflammation and TRD in both humans and animals in the same study. Clinical and experimental studies have been conducted for several reasons. First, TRD cannot be accurately measured in humans. This was overcome by using an animal model of depression, where we measured inflammatory biomarkers in different treatment groups to investigate the difference between them and rule out TRD. In addition, the relationship between TRD and known inflammatory biomarkers has been studied from both perspectives to confirm the clinical findings, which were assumed to demonstrate an association between inflammation and TRD. Second, this study aimed to determine the mechanism underlying the association between inflammation and TRD which cannot be studied in humans. Thus, the CUMS rat model of depression was used to examine the association between JAK/STAT and TRD signaling pathways, as no evidence indicating such a relationship exist in the literature. Several mechanisms involved in TRD, including inflammation, oxidative stress and the hypothalamic–pituitary–adrenal (HPA) axis, have been described [39,40]. This study clearly demonstrated the relationship between inflammation and TRD in a rat model of depression by targeting the JAK/STAT signaling pathway. Third, understanding the link between inflammation and TRD will lead to new treatments for managing depression, as measuring the expression levels of CRP and other cytokines allows for individualizing treatment plans for patients. Additionally, measuring inflammatory biomarkers and targeting inflammation or their mediators may be appropriate for patients with depression who have undergone multiple antidepressant treatment trials, on the basis of evidence indicating that the measurement of baseline inflammatory biomarkers could be useful predictors of treatment response in MDD [41,42]. New drugs can be developed exclusively for patients with TRD due to inflammation. Finally, new guidelines and therapeutic algorithms should be designed for patients with depression and resistance to antidepressants to help them achieve their treatment goals. Determining the link between inflammation and TRD positively affects the health status and quality of life of patients with depression.

The clinical results in the first analysis confirmed the study hypothesis. The number of inflammatory biomarkers (CRP, ESR and WBC count) was significantly higher in patients with MDD than in the healthy volunteers. Moreover, the results of the second analysis showed that the treatment-resistant group tested positive for more inflammatory biomarkers than the treatment responders. The results of both analyses were consistent and supported the hypothesis that increased levels of inflammatory biomarkers are risk factors for TRD. These findings provide strong evidence linking inflammation to TRD.

Cortisol is another important stress marker that regulates various physiological, emotional, and cognitive processes. It is predicted to be involved in the etiology of MDD, as its regulation is considered important in the treatment and remission of major depression [43]. However, owing to missing data and laboratory results in the patient's medical records, cortisol levels could not be analyzed. Thus, stress biomarkers were examined in an experimental study using an animal model of depression. Further clinical studies on the influence of cortisol on TRD are needed in the future.

In the experimental method, FLX was used as a preventive antidepressant, LPS was used as an inflammatory agent, the CUMS protocol was used for inducing depression-like behavior, and behavioral tests (SPT and FST) were used to assess stress, in accordance with previous studies [21,44]. FLX was chosen for this study because it is the most widely used medication worldwide, owing to its safety profile [45,46]. A previous study examining the preventive role of FLX has been utilized. However, the small sample size limited the evidence [47].

Numerous studies have proven the ability of LPS to mimic inflammation. The expressions of pro-inflammatory cytokines (IL-1$\beta$, IL-6 and TNF-$\alpha$) were induced by peripheral administration of LPS in rats and were found in the plasma of patients with TRD [48–50]. This study showed that LPS administration potentially triggered inflammation and consequently attenuated the antidepressant effect of FLX. The CUMS protocol was used in this experiment because of its known reliability and effectiveness in inducing depression-like behavior in rats and its great validity and translational potential in humans [51–53]. In the experiments, the animals were exposed to several unpredictable stressors daily to enhance the relevance of this model to humans. Behavioral tests (FST and SPT) were performed to assess the effects of the CUMS protocol and the actions of antidepressant treatments. These tests provide validity to support the interpretation of depression-like behaviors and are the gold standard tests for assessing despair. The SPT reflects face validity [54,55] for detecting disorders, and the FST reflects validity for predicting the efficacy of antidepressant treatment action [54,55].

The experimental results were similar to those of a previous study in which exposure to LPS clearly reversed the antidepressant action of FLX and led to TRD [21]. In the behavioral tests, the SPT was used to assess anhedonia, and the FST was used to measure immobility time. This study showed that stress exposure decreased the rats' preference for the sucrose solution, which indicated anhedonia, and that stress increased the immobility time. Treating stressed rats with FLX for 4 weeks increased anhedonia and mobility time. However, LPS treatment inhibited anhedonia. Immobility time was increased because the inflammatory effect of LPS concealed the antidepressant effect of FLX, thus reflecting resistance. In summary, the behavioral results clearly indicate that LPS mitigated the antidepressant effect of FLX, which led to TRD. This suggests a connection between the mechanisms of inflammatory action and the resulting behavioral changes.

The clinical findings of the animal study confirmed the results of the human study; the concentrations of inflammatory biomarkers (IL-6, TNF-$\alpha$ and CRP) and stress biomarker (corticosterone) were higher in the CUMS group not treated with FLX and exposed to LPS. The most significant increase in the levels of biomarkers were observed in the FLX-treated CUMS and LPS groups, as the inflammatory biomarkers prevented FLX from functioning as an SSRI [21]. One most supported theory to explain the role of inflammation in TRD proposes that increased levels of inflammatory cytokines lead to the activation of the HPA axis and failure of negative feedback loops, which promotes TRD [39]. LPS clearly inhibited the antidepressant effects of FLX, which suggests cellular and molecular mechanisms that link inflammation to TRD.

The JAK/STAT pathway is also involved in the development and progression of mood disorders and may be involved in the pathophysiology of depression [15]. However, no evidence has been reported on the mechanism that links JAK/STAT signaling pathways to TRD until recently. Thus, the JAK2/STAT3 signaling pathway was targeted to evaluate its crucial role in immune and inflammatory responses, as its dysregulation is a key factor in

various neurodegenerative diseases [56]. JAK2 and STAT3 are the most highly expressed of the four JAK isoforms (JAK1, JAK2, JAK3 and TYK2) and seven STAT isoforms (STAT1, STAT2, STAT3, STAT5A, STAT5B and STAT6) found in the post-synaptic density in the brain. The activation of the JAK2/STAT3 pathway is critical for the rapid induction of long-term depression in hippocampal synapses [56,57]. IL-6 triggers the JAK/STAT3 signaling cascade and modulates activation of the HPA axis in depression; thus, increasing IL-6 levels contributes to depression. In the present study, the gene expression was detected in the hippocampus. This underlying neurogenesis-dependent mechanism shows anti-FLX activity-related inflammation and provide strong evidence linking inflammation to TRD.

This study demonstrates the link between inflammation and TRD in both clinical human and animal models and thus could pave the way for further studies that target the downstream effects of inflammation on the JAK/STAT signaling pathway. Thus, anti-inflammatory drugs may be a new option in the treatment of patients with TRD. Furthermore, according to the proven relationship, we recommend that clinicians measure the levels of CRP and other inflammatory and stress biomarkers before and during treatment. This may be useful in choosing and individualizing treatment regimens, as these indicators may provide clues regarding the physiological response of patients to antidepressants. Algorithms may be designed specifically for patients with TRD, and new guidelines may be established and applied.

*Study Limitations*

We focused our preclinical studies on male rats due to gender differences in pharmacokinetics and pharmacodynamics of antidepressants, which suggest that males respond differently to a treatment. Also, the presence of estrogen in females of reproductive age may interfere with the mechanism of action of multiple antidepressant drugs [37,38]. Another point is that our observation regarding JAK/STAT is preliminary, and it is for future studies to characterize this signaling at proteomics and mRNA levels. The comparison of histological data was based on the representative slides of brain sections from five rats per group, but provided inadequate information from the quantification of histological data and statistical comparison between the groups. Thus, our future direction is to investigate whether the attenuation of the normalization and preventative effects of FLX is dependent on the JAK/STAT signaling cascade by targeting the inhibition of the JAK/STAT pathway. This will largely depend on the quantification of histological and immunohistochemical data, for which statistical analysis is essential.

## 5. Conclusions

This study highlights that chronic stress and high inflammation hinder the antidepressant effects of FLX, and that the link between inflammation and TRD was clearly attributed to the dysregulation of the JAK2/STAT3 signaling pathway. The increased gene expression of the phosphorylated JAK2/STAT3 proteins in the brain hippocampus of the FLX-treated (CUMS + LPS) rats confirms the anti-FLX effect of inflammation. The clinical study results confirmed that inflammation is a clinical component of depression, supporting the existing reports. Therefore, some patients' failure to respond to antidepressant treatments could be mitigated by treating inflammation, which might improve the antidepressant effect in patients with TRD. Moreover, identifying the link between inflammation and TRD might lead to the development of a novel medication for the treatment of chronically ill patients with TRD.

**Supplementary Materials:** The following supporting information can be downloaded at: https://www.mdpi.com/article/10.3390/neurolint15010009/s1, Table S1: title: Descriptive statistics for study sample; Table S2: title: Comparison of the treatment respondent group vs. treatment-resistant group in the demographic and clinical data. Table S3: Comparison of the Laboratory Test Results between the Healthy Group vs. Depressed Group; Table S4: Comparison of the Laboratory Test Results between the Respondent Group vs. Resistant Group; Figure S1: Experimental Protocol.

**Author Contributions:** L.F.A. Conceptualization. L.F.A. and R.A.A. performed all the clinical and experimental work. They collected the data, performed all experiments, analyzed, and interpreted the data, designed the figures and wrote the manuscript. J.F.A. contributed to the general guidance of the clinical study and the review of the manuscript. A.M.A. contributed to clinical data collection and review of the manuscript. The M.A.A. provided support for the study by helping to perform all experiments and experimental analyses. H.M.A. contributed to the experimental results. A.S.A. helped generate graphs with high resolution using the GraphPad Prism software and reviewed the manuscript. A.M.E.-M. helped in clinical data analysis and representation. N.M.A. supervised and administered the project, critically reviewed and edited the manuscript. All authors have read and agreed to the published version of the manuscript.

**Funding:** This research received no funding.

**Institutional Review Board Statement:** All the clinical study procedures were approved by the research ethics committee (National Committee of Bioethics, Ethics Service and the Institutional Review Board of King Saud University (KSU-IRB); approval No. E-20-4971 (07/12/2020)). All the animal study protocols and experiments were approved and conducted in accordance with the Experimental Animals Ethics Committee Act of King Saud University, Institutional Research Ethics Committee (Ethics Reference No. KSU-SE-20-23 (0805/2020)).

**Informed Consent Statement:** Patient consent was waived due to reason; as it is a retrospective cohort study, the patients were enrolled by reviewing their medical records only and, by the approval of the research ethics committee, this gives us the right to collect the data solely for the purpose of this study and the patient's information were confidential.

**Data Availability Statement:** The data presented in this study are available on request from the corresponding author.

**Acknowledgments:** We thank Ahmed El-Malky, the Morbidity and Mortality Review Unit Deputy Supervisor at King Saud University Medical City, for his support in clinical data collection and clinical data analysis. Moreover, we thank Khalid Elfakki Abdullah, Department of Zoology, College of Science at King Saud University, for his support with the histology and immunohistochemistry of the experimental results.

**Conflicts of Interest:** The authors declare that they have no conflict of interest or personal relationships that could have influenced the work reported in this paper.

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
