# Peer review of "Inflammation and Treatment-Resistant Depression from Clinical to Animal Study: A Possible Link?"

_2035-8377, doi:10.3390/neurolint15010009_

Round 1

Reviewer 1 Report

This very interesting manuscript is devoted to investigating the relationship between treatment-resistant depression (TRD) and inflammation in humans and experimental models. Several hypotheses were tested to confirm the association between inflammation and TRD.

This study was conducted with a strong belief that proving the link between inflammation and TRD in both clinical human and animal models could pave the way for further studies that target the downstream effects of inflammation on the JAK/STAT signaling pathway, which can be an important therapeutic option for patients with TRD. Anti-inflammatory drugs may be a new option in the treatment of patients with TRD.

The experiment schemes are very clear and detailed. Figures 5 and 6 do not show statistical significant differences between groups.

Author Response

Reviewer#1

The experiment schemes are very clear and detailed. Figures 5 and 6 do not show statistical significant differences between groups.

Response:

Thank you for your consideration, Figure 3 (Note that formerly named figure 5) and Figure 3 (Note that formerly named figure 6) are adjusted accordingly, and the statistical significance are added on the figures. Kindly find the revised figures, we added them to the manuscript.

Reviewer 2 Report

This is a potentially interesting paper, which includes complementary clinical and preclinical studies, but there are a number of problems that decrease my enthusiasm.

The major issues are:

1. There is considerable lack of clarity about the clinical study, including uncertainty about the provenance of the control group, a lack of explanation of the differential diagnosis of antidepressant responsive and non-responsive groups, and large volumes of missing data.

2. There are discrepancies in the data between text and figures, and if the figures are correctly described, some of the results presented as significant are notl.

3. The results are largely confirmatory of published data, and it is not immediately apparent what is the novel contribution. I think it is the histological data providing evidence for involvement of the JAK-STAT pathway in antidepressant resistance. However, these slides represent n=1 per group, and the number of brains examined is not stated. I think these results need to be quantified with statistical analysis, and this may require repetition of the study to increase the sample sizes.

4. The Results section is poorly written, with repetition in numerical form of all of the data shown in the figures.

Specific comments:

Introduction

5. “Numerous clinical factors that trigger TRD lead to inflammation.” This sentence is not followed by any evidence about factors that lead to TRD

6. “Generally, patients with increased levels of inflammatory markers at baseline have been found to be less responsive to antidepressant medications [6,13].” It needs to be made clear how the clinical part of the present study extends these earlier observations. What is novel about the present study?

Methods

7. I don’t understand where the healthy volunteers come from, given that the inclusion criteria were “patients aged 18 years, male or female, with an Axis-1 115 clinical diagnosis of MDD and taking antidepressants”. 

8. The description of the methodology for identifying treatment resistance is unclear. How exactly was treatment resistance determined from the factors listed? How were missing data dealt with?

9. Figure 1 adds nothing and should be deleted.

10. Why were only male rats used?

11. The design of the experimental study is missing a control group treated with LPS and FLX. With that group we would have a balanced design that would be analyzed by 3-way anova (with/without stress/LPS/FLX). Without it we do not, so the analysis is sub-optimal.

12. FLX was administered alongside LPS/CMS. This is preventative treatment. It is much less relevant to TRD than a procedure in which drug treatment is given only after the behavioral impairments have been established.

13. The procedure for the SPT states “Second, the rats were housed individually”.             Earlier it is stated that rats were housed individually throughout the experiment. Please clarify.

14. Fig. 3 adds nothing. What would be informative is a timeline showing the FST and SPT procedures in the same figure.

15. Why the focus on the hippocampus?

16. How many rats from each group were selected for histopathological analysis?

17. The appropriate analysis for the sucrose data is 2-way anova.

Results

18. Again, I don’t understand where the healthy control group came from, given all participants were depressed and on medication.

19. There is no explanation of the missing 24 patients (103-79) in the depressed group 

20. The abstract is misleading in stating that there were 650 participants.  650 were screened but only 206 participated.

21. Tables 1 and 2 are misplaced. The first analyses are for healthy vs. depressed. These tables which compared two groups of depressed patients should come later.

22. The comparison of healthy vs. depressed on education is invalid – Most of these data are missing.

23. I cannot evaluate the clinical data because I do not understand the basis for designating patients as responsive or resistant. I am also concerned about the very low numbers in some of these analyses.

24. Some of the errors reported in the text are different from those shown in Fig. 5. And if these are standard errors, as reported in the Methods, then the ESR and WBC data are far from being significant. There is something badly wrong here.

25. Exactly the same – on both points – is true of the comparison between the two depressed groups.

26. The legends to figures should not repeat the numbers shown in the Figure and already cited in the text. 

27. The sucrose data are problematic because there were substantial variations in sucrose consumption at baseline. In particular, baseline consumption in the CUMS group was low, with the result that there was no significant effect of CUMS (relative to baseline).  The baselines in the LPS groups are also problematic. It is clear that FLX was ineffective in the LPS+CMS group. However, there is also no significant effect of FLX relative to CUMS when the baseline differences are taken into account.

28. The FST, corticosterone and inflammatory marker results are clearer. As noted earlier, it is unfortunate that a group administered LPS and FXT without CUMS was omitted.

29. To my eye, the histology slides (Fig. 11) all look the same, other than panel B (CUMS). That is, FXT was effective against CMS, but there was no obvious damage in the LPS or LPS+CMS groups, or LPS+CMS+FXT. I do wonder about the reliability of these data, being based on a tiny sample. (ie. subsamples from an initial n=5). The inflammatory markers (Fig.12) look more convincing, but again, I am concerned about the sample sizes for these comparisons. And the word “significant” should certainly not be used in relation to these data.

Discussion

30. I am not sure that examining the role of inflammation “in both humans and animals in the same study” is a particular virtue, given how different the two halves of this study were. 

31. Examining the data presented, the clinical study confirms what is already known, that there is an association between inflammation and treatment resistance in depression (assuming that the results are reliable, which, as detailed above, is uncertain from the paper as it stands). The value of the experimental study rests very largely on the histological data shown in Fig. 12 (since the lack of effect of FXT against LPS is already known), which lacks information about the sample sizes on which the comparisons are based and the representativeness of the slides selected for presentation. As it stands, these slides represent n=1 per group. If, as I suspect, this is the main point of the paper, then quantification and statistical analysis are essential.

32. If attention to the issues raised earlier leads to changes in the results reported, this would feed through to corresponding changes in the Discussion.

Author Response

Reviewer#2

Reviewer comments:

The major issues are:

  1. There is considerable lack of clarity about the clinical study, including uncertainty about the provenance of the control group, a lack of explanation of the differential diagnosis of antidepressant responsive and non-responsive groups, and large volumes of missing data.
  2. There are discrepancies in the data between text and figures, and if the figures are correctly described, some of the results presented as significant are notl.
  3. The results are largely confirmatory of published data, and it is not immediately apparent what is the novel contribution. I think it is the histological data providing evidence for involvement of the JAK-STAT pathway in antidepressant resistance. However, these slides represent n=1 per group, and the number of brains examined is not stated. I think these results need to be quantified with statistical analysis, and this may require repetition of the study to increase the sample sizes.
  4. The Results section is poorly written, with repetition in numerical form of all of the data shown in the figures.

Thank you for your constructive comment. Below are our responses for each point.

Specific comments:

Introduction

  1. “Numerous clinical factors that trigger TRD lead to inflammation.” This sentence is not followed by any evidence about factors that lead to TRD.

Modifieed to:

Numerous clinical factors linked TRD to inflammation.

Besides, the reference of (Raison, Felger, & Miller, 2013; Strawbridge et al., 2015) stated that the clinical factors and comorbid conditions associated with inflammation, such as childhood trauma, obesity, and medical illness, are associated with treatment-resistant depressive episodes.

Raison, C.L.; Felger, J.C.; Miller, A.H. Inflammation and Treatment Resistance in Major Depression: The Perfect Storm. Psychiatr. Times 2013, 30, doi:10.1016/j.bbi.2011.07.214.

  1. “Generally, patients with increased levels of inflammatory markers at baseline have been found to be less responsive to antidepressant medications [6,13].” It needs to be made clear how the clinical part of the present study extends these earlier observations. What is novel about the present study?

Initially, according to a special issue of the journal Science, the recognition that inflammation is a primary pathophysiologic mechanism in chronic illness is one of the major scientific insights of the decade (Science, Volume 330, issue 6011, page 1621, 2010). Some of the earliest evidence that major depression was associated with increased inflammation came from the work of Maes et al (1991, 1992a, b). Subsequently, increased inflammation in depression appears to be a state-dependent phenomenon, with a number of studies demonstrating that increased inflammatory markers in depressed patients return to control levels following successful antidepressant treatment (Miller et al, 2009). A corollary to these findings is that depressed patients who fail to respond to antidepressant therapy show increased inflammatory markers (Sluzewska et al, 1997; Lanquillon et al, 2000; Fitzgerald et al, 2006). Moreover, patients with increased inflammatory markers at baseline are less likely to show a response to treatment, suggesting a relationship between inflammation and treat- ment resistance (Sluzewska et al, 1997; Lanquillon et al, 2000; Fitzgerald et al, 2006). The clinical part of this study extends to all of these earlier observations and approved such a relationship  as it examines directly the relation between the increased levels of inflammatory biomarkers and depression treatment non-response  via retrospective cohort study in humans model and empowering the clinical findings by studying this relation in an  animal model of depression  which allowed us to take a step forward in the investigating the underlying pathophysiology.

Citation:  Haroon,  E.;  Raison,  C.L.;  Miller,  A.H.  Psychoneuroimmunology  Meets  Neuropsychopharmacology:  Translational   Implications  of  the  Impact  of  Inflammation  on  Behavior.  Neuropsychopharmacology  2012,  37,  137–162,  doi:10.1038/npp.2011.205.

Methods

  1. I don’t understand where the healthy volunteers come from, given that the inclusion criteria were “patients aged 18 years, male or female, with an Axis-1 115 clinical diagnosis of MDD and taking antidepressants”. 

Healthy volunteers criteria:

  • The ones with no history of Axis-1 115 clinical diagnosis of MDD.
  • No underlying systematic inflammatory comorbid illness (rheumatoid arthritis M05 all, M06 other rheumatoid arthritis ALL, M07 psoriasis ALL, Lupus L93)
  • No history of physical illnesses (diabetes mellitus (E10-E14), family history of stroke (Z82.3), family history of diabetes (Z83.3), ischaemic heart disease (Z82.4), Angina (I20.0-I20.9), myocardial infarction (I21.0 to I21.9), subsequent myocardial infarction (I22.0 to I22.9), atherosclerotic heart disease (I25.0 to I25.9), family history of mental and behavioral disorders (Z81.0 and Z81.8)
  • Had no malignant neoplasm C00 to C97), malignant neoplasms C76 to C80 ALL, and personal history of malignant neoplasm (Z85).
  • Had no Manic (F30), bipolar (F31), schizophrenia (F320-F29) disorders.
  • Had no (F10-F19) mental and behavioral disorders due to psychoactive substance use.
  • No Pregnancy or breastfeeding in the past 6 months
  • Had no history of taking antidepressant treatment or chronic anti-inflammatory medication.

Kindly find below the main references that we referred to in the methodology:

Chamberlain, S. R., Cavanagh, J., de Boer, P., Mondelli, V., Jones, D. N. C., Drevets, W. C., Cowen, P. J., Harrison, N. A., Pointon, L., Pariante, C. M., & Bullmore, E. T. (2019). Treatment-resistant depression and peripheral C-reactive protein. The British journal of psychiatry : the journal of mental science, 214(1), 11–19. https://doi.org/10.1192/bjp.2018.66

Bell, J. A., Kivimäki, M., Bullmore, E. T., Steptoe, A., MRC ImmunoPsychiatry Consortium, & Carvalho, L. A. (2017). Repeated exposure to systemic inflammation and risk of new depressive symptoms among older adults. Translational psychiatry, 7(8), e1208. https://doi.org/10.1038/tp.2017.155

Zalli, A., Jovanova, O., Hoogendijk, W. J., Tiemeier, H., & Carvalho, L. A. (2016). Low-grade inflammation predicts persistence of depressive symptoms. Psychopharmacology, 233(9), 1669–1678. https://doi.org/10.1007/s00213-015-3919-9

  1. The description of the methodology for identifying treatment resistance is unclear. How exactly was treatment resistance determined from the factors listed? How were missing data dealt with?

The reference that was used in identifying the factors is: Dawn F. Ionescu J. Pharmacological approaches to the challenge of treatment-resistant depression [Internet]. PubMed Central (PMC). 2020 [cited 15 March 2020]. Available from: https://www.ncbi.nlm.nih.gov/pmc/articles/PMC4518696/.

First we extracted these data from each patient file, we looked at the patient history as it’s usually mentioned in the file, and by looking through the patient medication list, previous medications used and the doctor document viewing we extracted if any changes in antidepressant medications happened in  the  current  year, we determined the  number  of  previous  depressive  episodes and the length of each episode,  and we looked at the patient number of visits since diagnosis of depression, and did he tried to commit  suicide .. all of these information was found in the patient file. When we faced any missing data that we couldn’t figure it out, we referred directly to the patient or one of his family members by calling them from the clinic with the help of their doctor. All work was done professionally to achieve the maximal reliable results.

  1. Figure 1 adds nothing and should be deleted.

This figure represents the animal grouping and can facilitate tracking the study’s groups for the study readers.

  1. Why were only male rats used?

We understand the need to include females in the study since sexual dimorphism does exist in brain disorders, including mood disorders. However, we excluded females from our settings due to the profound hormonal impact, particularly the estrous cycle fluctuation. It would be good to include it in future studies. 

  1. The design of the experimental study is missing a control group treated with LPS and FLX. With that group we would have a balanced design that would be analyzed by 3-way anova (with/without stress/LPS/FLX). Without it we do not, so the analysis is sub-optimal.

Thank you for your note. Indeed it would be ideal- due to the significant number of animal groups we utilized with the large overall number we used. We believed in minimizing the animal used for our experimental purposes to address our research question adequately. We already have a FLX in environmentally triggered depression (Gp4), and FLX in typical housing conditions (Gp3). Similar utilization for LPS (Gp5 and 6).

Besides, a previous report has utilized LPS+FLX, and from their report, we found that this group is not statistically different from the control group.Ref: Xanthohumol Attenuates Lipopolysaccharide-Induced Depressive Like Behavior in Mice: Involvement of NF-κB/Nrf2 Signaling Pathways.

Link: https://link.springer.com/article/10.1007/s11064-021-03396-w

  1. FLX was administered alongside LPS/CMS. This is preventative treatment. It is much less relevant to TRD than a procedure in which drug treatment is given only after the behavioral impairments have been established.

Our central goals have been adjusted line (83-87),

we examined the preventive role of FLX in environmental and pharmacologically relevant models of depression. Accordingly, we hypothesized that inflammation contributes to TRD. In addition, we observed a potential involvement of the JAK/STAT signaling pathway in our preclinical study settings.

  1. The procedure for the SPT states “Second, the rats were housed individually”.             Earlier it is stated that rats were housed individually throughout the experiment. Please clarify.

First, we added the two bottles (water and sucrose), On the second day, we placed each rat individually, and later we calculated the amount of sucrose consumed.

  1. Fig. 3 adds nothing. What would be informative is a timeline showing the FST and SPT procedures in the same figure.

Addressed.

  1. Why the focus on the hippocampus?

We choose this area because the hippocampus is comprehensively studied in the neuroscience field. Besides the assembly of the hippocampal structure is simple and can be identified and extracted easily. Likewise, alterations in the hippocampal formation are implicated in the pathology of several brain diseases(Andersen et al. 2009; Winterer et al. 2011). Most importantly, atrophy in the hippocampal area is linked to depression and chronic stress(Andrus et al. 2012).

  1. How many rats from each group were selected for histopathological analysis?

These are representative images of brains of five rats per group we didn't analyse them, modifications have been implemented.

  1. The appropriate analysis for the sucrose data is 2-way anova.

Thank you for the constructive comment. In accordance with your recommendation, we have deleted the basal data and re-analyze the final data using one-way ANOVA. As our focus was on the preventive role of fluoxetine (Flex), we only included the sucrose consumption percentage after stress. The result in G7 (Flex-treated LPS plus CUMS group) showed that Flex displayed no normalization or protective effect under environmentally and pharmacologically stressful conditions.

  1. Again, I don’t understand where the healthy control group came from, given all participants were depressed and on medication.

We mentioned this in the method section line(114-125), probably because the tables’ presentation order this information was unclear, so we re-arranged our table presentation and moved some to the supplementary.

The inclusion criteria were patients aged 18 years, male or female, with an Axis-1 clinical diagnosis of MDD and taking antidepressants. The patients were divided into two groups for analyses, with each analysis further divided into two groups, either (1) healthy volunteers vs. patients with depression or (2) treatment responders vs. non-responders. The patient exclusion criteria were as follows: (1) patients diagnosed with Axis-1 psychiatric disorders other than depression, such as schizophrenia or bipolar disorder; (2) pregnant or breastfeeding during the study period; (3) taking medication that inhibited the immune system, such as corticosteroids and non-steroidal anti-inflammatory drugs; and (4) patients with a known history of alcoholism or drug abuse. In the sensitivity analysis, we also excluded patients with an acute infection based on the possibility of extremely skewed CRP and WBC levels at the time of blood collection [13,15].

  1. There is no explanation of the missing 24 patients (103-79) in the depressed group 

Highly appreciating your accuracy, yes 79 patients had MDD (39 Respondents and 40 Resistant) The remaining 24 patients were "responders" also but we had to exclude them from the flow chart presentation because they were classified as "acquiescence" so they could be  conceptualized as a reluctant acceptance of participation without protest.   

  1. The abstract is misleading in stating that there were 650 participants.  650 were screened but only 206 participated.

adjusted.

  1. Tables 1 and 2 are misplaced. The first analyses are for healthy vs. depressed. These tables which compared two groups of depressed patients should come later.

Thank you for your constructive comment, adjusted.

  1. The comparison of healthy vs. depressed on education is invalid – Most of these data are missing.

we re-arranged our table presentation and moved some to the supplementary. The highlighted results were kept in the original manuscript.

  1. I cannot evaluate the clinical data because I do not understand the basis for designating patients as responsive or resistant. I am also concerned about the very low numbers in some of these analyses.

As mentioned above : we re-arranged our table presentation and moved some to the supplementary. The highlighted results were kept in the original manuscript.

  1. Some of the errors reported in the text are different from those shown in Fig. 5. And if these are standard errors, as reported in the Methods, then the ESR and WBC data are far from being significant. There is something badly wrong here.

Adjusted (Note that Fig 5 is now named Fig3). Kindly refer to Q25 below for answer.

  1. Exactly the same – on both points – is true of the comparison between the two depressed groups.

Response to comment 24 and 25: What was mentioned in the tables, was the “standard deviation” and accordingly the CI and P value were calculated. As mentioned in the table, the results were statistically significant. However, we plotted on the graphs, the results and labeled values with the errors bars according to the request of one of the journal reviewers.

  1. The legends to figures should not repeat the numbers shown in the Figure and already cited in the text.

 Thank you for notification and we adjusted them.

  1. The sucrose data are problematic because there were substantial variations in sucrose consumption at baseline. In particular, baseline consumption in the CUMS group was low, with the result that there was no significant effect of CUMS (relative to baseline).  The baselines in the LPS groups are also problematic. It is clear that FLX was ineffective in the LPS+CMS group. However, there is also no significant effect of FLX relative to CUMS when the baseline differences are taken into account.

Thank you for the constructive comment. In accordance with your recommendation, we have deleted the basal data and reanalyze the final data using one-way ANOVA. As our focus was on the preventive role of fluoxetine (Flex), we only included the sucrose consumption percentage after stress. The result in G7 (Flex-treated LPS plus CUMS group) showed that Flex displayed no normalization or protective effect under environmentally and pharmacologically stressful conditions.

  1. The FST, corticosterone and inflammatory marker results are clearer. As noted earlier, it is unfortunate that a group administered LPS and FXT without CUMS was omitted.

The same response to reviewer`s comment#11 "Thank you for your note. Indeed it would be ideal- due to the significant number of animal groups we utilized with the large overall number we used. We believed in minimizing the animal used for our experimental purposes to address our research question adequately. We already have a FLX in environmentally triggered depression (Gp4), and  FLX in typical housing conditions (Gp3). Similar utilization for LPS (Gp5 and 6).

Besides, a previous report has utilized LPS+FLX, and from their report, we found that this group is not statistically different from the control group.Ref: Xanthohumol Attenuates Lipopolysaccharide-Induced Depressive Like Behavior in Mice: Involvement of NF-κB/Nrf2 Signaling Pathways".

Link: https://link.springer.com/article/10.1007/s11064-021-03396-w

  1. To my eye, the histology slides (Fig. 11) all look the same, other than panel B (CUMS). That is, FXT was effective against CMS, but there was no obvious damage in the LPS or LPS+CMS groups, or LPS+CMS+FXT. I do wonder about the reliability of these data, being based on a tiny sample. (ie. subsamples from an initial n=5). The inflammatory markers (Fig.12) look more convincing, but again, I am concerned about the sample sizes for these comparisons. And the word “significant” should certainly not be used in relation to these data.

 Thank you for notification and we addressed them in a study limitation section, lines (750-757).

Discussion

  1. I am not sure that examining the role of inflammation “in both humans and animals in the same study” is a particular virtue, given how different the two halves of this study were.

Yes they are different but they do complete each other. As you know TRD is not a measurable variable that can be detected by a certain device as blood pressure for example, so no matter how the clinical study was strong the results will never be 100% definitive. By using the animal study any difficulty was faced in the clinical studying it was overcome by using rat model of depression (we used this model for studying according the previous studies and how they nearly match with humans) extra to that the animal model didn’t just give us and insight in the link between inflammation and TRD by looking at the inflammatory biomarkers effect in the different groups but also helped us it investigating in the underlying mechanism which was couldn’t be done in humans study, that’s how they complete each other.

  1. Examining the data presented, the clinical study confirms what is already known, that there is an association between inflammation and treatment resistance in depression (assuming that the results are reliable, which, as detailed above, is uncertain from the paper as it stands). The value of the experimental study rests very largely on the histological data shown in Fig. 12 (since the lack of effect of FXT against LPS is already known), which lacks information about the sample sizes on which the comparisons are based and the representativeness of the slides selected for presentation. As it stands, these slides represent n=1 per group. If, as I suspect, this is the main point of the paper, then quantification and statistical analysis are essential.

Thank you for notification and we addressed them in a study limitation section, lines (757-765).

  1. If attention to the issues raised earlier leads to changes in the results reported, this would feed through to corresponding changes in the Discussion.

Noted – adjusted we added:

a study limitation paragraph line (750-757): We focused on our preclinical studies on male rats due to gender differences in pharmacokinetics and pharmacodynamics of antidepressants, which suggest that males respond differently to a treatment. Also, the presence of estrogen in females of reproductive age may interfere with the mechanism of action of multiple antidepressant drugs [34,35]. Another point is that our observation regarding JAK/STAT  is preliminary, and it is for future studies to characterize this signaling at proteomics and mRNA level.

Another modified part  line (650-657): In this experimental method, FLX was used was used as a preventive antidepressant, LPS as an inflammatory agent, the CUMS protocol for inducing depression-like behavior and behavioral tests (SPT and FST) for assessing stress were chosen in accordance with previous studies [19,41]. FLX was chosen for this study because it is the most widely used medication worldwide due to its safety profile [42,43]. A previous study examining the preventive role of FLX has been utilized. However, the small sample size limited the evidence.

(reference: Prophylactic efficacy of fluoxetine, escitalopram, sertraline, paroxetine, and concomitant psychotherapy in major depressive disorder: Outcome after long-term follow-up

https://www.sciencedirect.com/science/article/abs/pii/S016517811400910X ).

Reviewer 3 Report

“Inflammation and treatment-resistant depression from clinical to animal study: a possible link?”

The aim of this manuscript by Lara F. Almutabagani et al. is to investigate the relationship between inflammation and treatment-resistant depression in both clinical and experimental setting. In addition, the authors provide observations of the possible link between specified signaling pathways, chronic stress and inflammation and the responsiveness to antidepressant.

The manuscript is about a current and major rising public health concern, due to the highly rising prevalence of depressive disorders worldwide and the need to improve the therapeutic strategies for the resistance to usual antidepressants. The study appears consistent with the recent literature and its strength is represented by the original focus on the role of inflammation and signaling pathways in the pathogenesis of resistant depression, especially in the context of a preclinical model. This study may inspirate the future research to deepen knowledge of baseline inflammatory biomarkers as useful predictor of treatment.

The introduction sets out the argument properly. The authors raise the medical issue providing epidemiological data. They illustrate several clinical factors that link TRD to inflammation, introducing the role of inflammatory markers and the singling pathways, involved in neurovegetative disease as well. They list the hypothesis underling the research, emphasizing the lack of the data available especially in preclinical study settings and the gaps in the current literature.

In the method section, the authors explain the methodology in dedicated paragraphs, “clinical study” and “experimental study”. All the procedures are declared to be approved by ethics committee, but it is not specified if the protocol study has been under approved registration. 

Each step of the methodological procedure is explained in detail, giving the idea of the precise structure of the study. Data are analyzed by a rich descriptive statistic and all the variables are well presented.

The “clinical study” consists in a retrospective cohort study conducted in Saudi Arabia from 2015 to 2020. The authors reviewed electronic medical records of the enrolled patients. The data source is appropriate for the sampling. Moreover, in the recruitment criteria, it is to notice that the “inflammatory biomarkers” selected in the study may be nonspecific to set out correlates of inflammation related resistant depression. Similarly, the investigation based on major chronic diseases or inflammatory illness may result too generic, with the risk of a wide target of patients. Could the authors provide a more detailed analysis of variance regarding the differences between the two MDD populations?

The “experimental study” is performed using male rats randomly assigned to stressed and non-stressed groups. The table simplify the understanding of the experimental protocol, which presents a very articulated configuration.

In the result and discussion sections, the authors maintain the previous structure of dedicated paragraph for both clinical and experimental studies. They describe the data findings, referring to the significance of the analyzed data. In the “clinical results”, it is not clear the exact number of the patients enrolled in the sample size, especially in the context of “Analysis two”, talking about “depressive patients” (n=103), “responder group” (n=39) and “resistant group” (n=40) (Where are the 24 missing patients?).

The discussion is too long and redundant at the expense of a good focus on the authors' translational links. A concise summarizing  and reorganization of this section is needed. Anyway, many flow and supplementary tables are provided in order synthetizes and catalog the findings. To completion, panels of histological brain sections are well presented to show differences between rats groups.

The list of references is adequate, but it would be appropriate referencing more recent research to emphasizes the originality of the study.

In the conclusions, the aims of the study seem to be achieved by the authors and the trends are plausible. However, despite the appreciable complexity of the study, it is difficult to define a clear parallelism between the clinical and the experimental part of the research. More explanation may be useful to give overall coherence.

Author Response

Comments and Suggestions for Author

“Inflammation and treatment-resistant depression from clinical to animal study: a possible link?”

The aim of this manuscript by Lara F. Almutabagani et al. is to investigate the relationship between inflammation and treatment-resistant depression in both clinical and experimental setting. In addition, the authors provide observations of the possible link between specified signaling pathways, chronic stress and inflammation and the responsiveness to antidepressant.

The manuscript is about a current and major rising public health concern, due to the highly rising prevalence of depressive disorders worldwide and the need to improve the therapeutic strategies for the resistance to usual antidepressants. The study appears consistent with the recent literature and its strength is represented by the original focus on the role of inflammation and signaling pathways in the pathogenesis of resistant depression, especially in the context of a preclinical model. This study may inspirate the future research to deepen knowledge of baseline inflammatory biomarkers as useful predictor of treatment.

Thank you and we highly appreciate your comment.

The introduction sets out the argument properly. The authors raise the medical issue providing epidemiological data. They illustrate several clinical factors that link TRD to inflammation, introducing the role of inflammatory markers and the singling pathways, involved in neurovegetative disease as well. They list the hypothesis underling the research, emphasizing the lack of the data available especially in preclinical study settings and the gaps in the current literature.

Thank you and we appreciate your consideration.

In the method section, the authors explain the methodology in dedicated paragraphs, “clinical study” and “experimental study”. All the procedures are declared to be approved by ethics committee, but it is not specified if the protocol study has been under approved registration.

Thank you for highlighting this issue for clarification. Our study is in compliance with the Saudi regulations for research ethics in animal and clinical studies such as retrospective study like our study. Our study design required approval from the local ethics committees: the study design for the clinical study was approved by the National Committee of Bioethics, Ethics Service and the Institutional Review Board of King Saud University under KSU-IRB approval No. E-20-4971; for the animal study, all experiments were conducted in accordance with the Experimental Animals Ethics Committee Act of King Saud University, and the study was approved by the Institutional Research Ethics Committee under Ethics Reference No. KSU-SE-20-23. however, the only studies that require approved registration in Saudi Clinical Trials Registry (SCTR) are the clinical trials (https://sctr.sfda.gov.sa), so our study does not require this type of approval.

Each step of the methodology procedure is explained in detail, giving the idea of the precise structure of the study. Data analyzed by a rich descriptive statistic and all the variables are well presented.

Thank you for this comment.

The “clinical study” consists in a retrospective cohort study conducted in Saudi Arabia from 2015 to 2020. The authors reviewed electronic medical records of the enrolled patients. The data source is appropriate for the sampling. Moreover, in the recruitment criteria, it is to notice that the “inflammatory biomarkers” selected in the study may be nonspecific to set out correlates of inflammation related resistant depression. Similarly, the investigation based on major chronic diseases or inflammatory illness may result too generic, with the risk of a wide target of patients. Could the authors provide a more detailed analysis of variance regarding the differences between the two MDD populations?

A significant   relationship was found between number of failed treatment trials and tumor  necrosis factor (TNF), CRP, WBC, 30–50%  of patients with depression are reported to have high  levels  of inflammatory markers, including the acute phase reactant           C-reactive protein       (CRP), as well as high body mass index (BMI), and/or  markers of metabolic dysregulation, all of    which  may predispose patients to the development of     co-morbid medical illnesses Inflammatory    cytokines that are produced in the periphery are known to access the central    nervous system           and to affect neurotransmitters and neural            circuits that contribute to the   behavioral symptoms of depression

Bekhbat, M., Chu, K., Le, N. A., Woolwine, B. J., Haroon, E., Miller, A. H., & Felger, J. C. (2018). Glucose and lipid-related biomarkers and the antidepressant response to infliximab in patients with treatment-resistant depression. Psychoneuroendocrinology, 98, 222–229. https://doi.org/10.1016/j.psyneuen.2018.09.004

Kindly find below more detailed analysis of variance regarding the differences between the two MDD populations

Multivariate analyses of covariance (MANCOVA) were calculated to test for main effects of group in each population and examine in more detail the analysis of variance regarding the differences between the two MDD populations, amounting to six separate MANCOVAs. As shown in Figure 1, the main effects of group were salient in all scenarios, and their effect sizes were all large (to ηp2, 0.14 was considered large).

Note that we also include a more detailed analysis of variance regarding the two MDD populations in the attached Supplementary File (Table S3 and Table S4).

The “experimental study” is performed using male rats randomly assigned to stressed and non-stressed groups. The table simplify the understanding of the experimental protocol, which presents a very articulated configuration.

We appreciate your suggestion. We think that experimental design (lines 220-240) concisely described the allocation of rats into different groups, and this information is also presented in Figure 1, which shows the experimental design. Another reviewer suggested that we delete three illustrations that presented the study protocol. Based on your comments, we will include these three previously deleted figures in the attached Supplementary file. If you still find that the experimental protocol is not clear, and you think we should include a simplified table, please let us know and we will revise manuscript accordingly. 

In the result and discussion sections, the authors maintain the previous structure of dedicated paragraph for both clinical and experimental studies. They describe the data findings, referring to the significance of the analyzed data. In the “clinical results”, it is not clear the exact number of the patients enrolled in the sample size, especially in the context of “Analysis two”, talking about “depressive patients” (n=103), “responder group” (n=39) and “resistant group” (n=40) (Where are the 24 missing patients?).

Thank you for your concern, to simplify it, Analysis one has a sample size which is wider than analysis 2

Enrolled patients:

  • Analysis 1 = (206 participants)
  • Analysis 2 = (79 participants)

The reason that analysis 1 had a wider sample of patients is that it’s easier to have the data of general depressed patients so we had (103 depressed patient) while differentiating between them into respondents and resistant patients we followed a long procedure -mentioned in the method-  .. the patients which couldn’t be ruled out with us into responders or resistance were excluded “as we wanted the data and the enrolled patients to be as precise as possible, any doubt we had regarding any patient was excluded , we can say that the 24 participants are responders but we couldn’t precisely classify these depressed patients as recovered patients and also we couldn’t classify them as non-responders, so these 24 patients are for surely depressed yes and since TRD is not a numerical measurable variable we had a doubt about their classification so we excluded them to have as much as possible precise results that reflects the truth possible correlation between inflammation and TRD.

In conclusion, the remaining 24 patients were "responders" also but we had to exclude them from the flow chart presentation because they were classified as "acquiescence" so they could be conceptualized as a reluctant acceptance of participation without protest.

The discussion is too long and redundant at the expense of a good focus on the authors' translational links. A concise summarizing and reorganization of this section is needed. Anyway, many flow and supplementary tables are provided in order synthetizes and catalog the findings. To completion, panels of histological brain sections are well presented to show differences between rats groups.

Thank for this comment. We have amended the discussion to make it shorter and reduce redundancy.

The list of references is adequate, but it would be appropriate referencing more recent research to emphasizes the originality of the study.

Thank you for your attention. We have updated and edited the references, accordingly, with changes highlighted in yellow.

In the conclusions, the aims of the study seem to be achieved by the authors and the trends are plausible. However, despite the appreciable complexity of the study, it is difficult to define a clear parallelism between the clinical and the experimental part of the research. More explanation may be useful to give overall coherence.

Thank you for this comment. We have addressed your concern in our conclusion.

Round 2

Reviewer 3 Report

The authors have satisfactorily resolved my concerns. The article is now, in my opinion, considered for publication.